# Voicebox: Text-Guided Multilingual Universal Speech Generation at Scale

**Matthew Le**[*]  **Apoorv Vyas**[*]  **Bowen Shi**[*]  **Brian Karrer**[*]  **Leda Sari**  **Rashel Moritz**

**Mary Williamson**  **Vimal Manohar**  **Yossi Adi**[†]  **Jay Mahadeokar**  **Wei-Ning Hsu**[*]

Fundamental AI Research (FAIR), Meta

## Abstract

Large-scale generative models such as GPT and DALL-E have revolutionized the research community. These models not only generate high fidelity outputs, but are also generalists which can solve tasks not explicitly taught. In contrast, speech generative models are still primitive in terms of scale and task generalization. In this paper, we present Voicebox, the most versatile text-guided generative model for speech at scale. Voicebox is a non-autoregressive flow-matching model trained to infill speech, given audio context and text, trained on over 50K hours of speech that are not filtered or enhanced. Similar to GPT, Voicebox can perform many different tasks through in-context learning, but is more flexible as it can also condition on future context. Voicebox can be used for mono or cross-lingual zero-shot text-to-speech synthesis, noise removal, content editing, style conversion, and diverse sample generation. In particular, Voicebox outperforms the state-of-the-art zero-shot TTS model VALL-E on both intelligibility (5.9% vs 1.9% word error rates) and audio similarity (0.580 vs 0.681) while being up to 20 times faster. Audio samples can be found in `https://voicebox.metademolab.com`.

## 1 Introduction

Recent advances in large-scale generative models [6, 42, 50] have led to a major paradigm shift towards building general-purpose models, which can perform many new tasks not explicitly trained on. These generative models learn to predict the missing data given the context. Post training, we can directly input a question, optionally with a few contextual question-answer examples, instead of fine-tuning with labeled data. While the training objective appears simple, it subsumes many tasks as one can convert them into some form of context. For the model to perform well at every task, it implies that the estimation of $p(\text{missing data} \mid \text{context})$ needs to be accurate for every context. Hence, scale and diversity are the most crucial factors for building general-purpose models [20, 1].

Despite the success of large-scale generative models in other areas, most speech models are still trained on datasets at the scale of tens to hundreds of hours [51, 31, 32, 46, 24, 58, 7]. Previous works consider highly curated datasets such as VCTK [64], which contains only clean audio recorded in studio from about 100 speakers with little speaking style and text variation. Such models struggle to synthesize speech with rich variation in emotion, voice, background noise, acoustic condition, and have not been tested on the abilities to generalize to tasks not explicitly trained on.

This paper presents Voicebox, the most versatile text-conditioned speech generative model at scale. Voicebox is trained on a text-guided speech infilling task, where the goal is to generate masked speech

---

[*]Equal contribution. Corresponding authors: `{mattle,wnhsu}@meta.com`
[†]FAIR & Hebrew University of Jerusalem.

37th Conference on Neural Information Processing Systems (NeurIPS 2023).

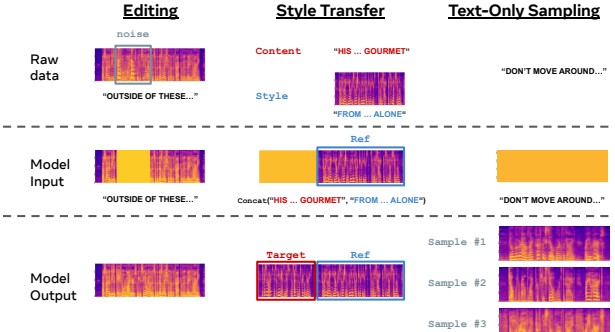

Figure 1: Task generalization via in-context learning.

Table 1: Comparing Voicebox with baselines on task capabilities. *Through infilling, A3T and Voicebox can remove transient noise but not stationary background noise.

| Model | ZS TTS | Denoise | Edit | Sampling |
|---|---|---|---|---|
| YourTTS | ✓ | ✗ | ✗ | ✓ |
| VALL-E | ✓ | ✗ | ✗ | ✓ |
| A3T | ✓ | * | ✓ | ✗ |
| Demucs | ✗ | ✓ | ✗ | ✗ |
| Voicebox | ✓ | * | ✓ | ✓ |

given its surrounding audio and text transcript. This can be considered as a guided in-context learning problem, where audio style is inferred from the audio context and textual content is specified through transcript. Voicebox does not require any audio style labels (e.g., speaker, emotion, and noise), which differentiates Voicebox from the majority of prior work where such labels are used extensively. Prior work uses labels to make the mapping between input (text and audio style) and output (speech) more deterministic to reduce underfitting [60, 46]. We show that Voicebox's text-guided speech infilling approach is much more scalable in terms of data while subsuming many common generative tasks.

In terms of modeling, Voicebox is a non-autoregressive (NAR) continuous normalizing flow (CNF) model [10]. Similar to diffusion models [19], CNFs model the transformation from a simple distribution to a complex data distribution, parameterized by a neural network. We train Voicebox with flow-matching [38], a recently proposed method that enables efficient and scalable training of CNFs via a simple vector field regression loss. In contrast to auto-regressive models, Voicebox can consume context not only in the past but also in the future. Moreover, the number of flow steps can be controlled at inference time to flexibly trade off quality and runtime efficiency.

Voicebox is trained on 60K hours of English audiobooks and 50K hours of multilingual audiobooks in 6 languages for the mono and multilingual setups. Voicebox achieves state-of-the-art performance on mono-lingual/cross-lingual zero-shot TTS, speech denoising, speech editing, diverse speech sampling and an application to data creation for speech recognition. To tackle the lack of comparability due to the use of subjective metrics, this paper presents a series of metrics using public models to facilitate reproducible comparison and model development for speech generation studies.

## 2 Related Work

**Generative speech models:** Most speech generative models are task-specific and trained on different datasets. One common type of task is *audio style conversion*, which aims to convert only a specific attribute while keeping other attributes the same. Voice conversion [27, 39], emotion conversion [53, 34], speech enhancement [63, 11, 55] belong to this category. Many of these models are supervised and trained on pairs of data that only differ in one attribute, for example, emotion [34]. It is hard to obtain such data. Moreover, some attributes, such as speaking style, are hard to annotate. Hence, these models are often trained on small datasets.

Controllable text-to-speech synthesis (TTS) is another common task, which aims to synthesize speech in a target audio style given text. While some styles like voice can be specified through labels [32] or pre-trained embeddings like YourTTS [7] and Jia et al. [25]; others like prosody are hard to annotate or embed. Previous studies [62] tried to control them by learning a residual embedding. However, these models encode style in a low-dimensional space and impose an overly simple distribution of speech given text and residual embedding [51, 56]. They cannot generate realistic noisy speech given a low dimensional vector, and performance degrades when conditioned on noisy references [21].

Infilling can be considered as another type of task. It aims to predict speech given context [36, 4] and optionally text guidance [3, 5, 61]. Instead of learning an explicit embedding to control style, infilling models predict speech coherent to the context. In other words, these models perform in-context learning similar to Large Language Models (LLM). While this is a step toward building large

scale generalist models using little explicit supervision, most prior work using text guidance still assumes a deterministic mapping from text and context to target [3, 5], which is only realistic for very short segments. Voicebox is a text-guided infilling model, but it leverages the CNF model that can parameterize any distribution. Hence, Voicebox can infill speech of any length and can be trained on in-the-wild datasets with rich variation, and provide a general solution that subsumes many tasks in a text-guided fashion.

**Large scale in-context learning models:** With the advancement in neural codec for speech [22, 12, 67], many recent studies explore token-based language modeling for speech generation. The GSLM-family [36, 28, 41] are textless language models built upon HuBERT units [22] for speech continuation without using text. HuBERT units encode mostly content, and the generated speech does not preserve the voice of the prompt. To tackle this, AudioLM [4] considers a cascaded approach which first generates HuBERT-like tokens and then predicts SoundStream [67] tokens, a reconstruction based codec that preserves style. These models are not conditioned on text and are evaluated on spoken language modeling tasks.

VALL-E [61] is most related to Voicebox. It is a text conditioned LM trained on Encodec [12] tokens (similar to SoundStream). Encodec encodes each frame with 8 ordered codebooks at 75Hz using a residual quantization layer. VALL-E has two modules. The first is an auto-regressive (AR) model that predicts the first code of each frame given text and the audio prompt. The second is an NAR model that predicts the remaining seven codebooks sequentially.

VALL-E demonstrates state-of-the-art (SOTA) zero-shot TTS performance through in-context learning, where speech of the desired style is used as prompt. The model considers the prompt as part of the whole utterance such that it generates the rest of the utterance containing the target text in the same audio style. Voicebox has several design advantages compared to this. 1) Voicebox can use context both in the past and future, which is useful for editing where only a segment in the middle needs to be generated. 2) Voicebox can generate speech much faster than VALL-E because flow-matching can produce high quality samples with less than 10 NAR steps, while VALL-E requires 1 AR and 7 NAR steps. 3) Voicebox decouples duration and audio modeling, enabling finer grained alignment control. 4) Voicebox is compatible with any continuous features including Encodec embeddings.

## 3 Method

### 3.1 Background: Flow Matching with an optimal transport path

Let $\mathbb{R}^d$ be the data space with data points $x \in \mathbb{R}^d$ drawn from some unknown distribution $q(x)$. Continuous Normalizing Flows (CNFs) [10] are a family of generative models that learn the transformation from a simple prior distribution $p_0$ (e.g., normal distribution) to the data distribution $p_1 \approx q$. CNFs parameterize a time-dependent vector field $v_t : [0, 1] \times \mathbb{R}^d \to \mathbb{R}^d$ that is used to construct a *flow*: $\phi_t : [0, 1] \times \mathbb{R}^d \to \mathbb{R}^d$ that pushes points from the prior towards the target distribution. The relationship is defined via the ordinary differential equation (ODE) as: $d\phi_t(x)/dt = v_t(\phi_t(x))$ and $\phi_0(x) = x$. For a flow $\phi_t$, the *probability path* (time-dependent probability density function) $p : [0, 1] \times \mathbb{R}^d \to \mathbb{R}_{>0}$ can be derived via the change of variables formula: $p_t(x) = p_0(\phi_t^{-1}(x)) \det \left[ \partial \phi_t^{-1}(x) / \partial x \right]$. To sample from $p_t(x)$, we first draw $x_0$ from $p_0$ and then solve the initial value problem (IVP) for $\phi_t(x_0)$ given $d\phi_t(x)/dt = v_t(\phi_t(x))$ and $\phi_0(x) = x_0$. We use $x_t$ and $\phi_t(x_0)$ interchangeably.

Let $p_t$ be a probability path and $u_t$ be the corresponding vector field that generates $p_t$. The vector field $v_t(x; \theta)$ parameterized by a neural network $\theta$ can be trained with the Flow Matching objective: $\mathcal{L}_{FM}(\theta) = \mathbb{E}_{t, p_t(x)} ||u_t(x) - v_t(x; \theta)||^2$, where $t \sim \mathcal{U}[0, 1]$ and $x \sim p_t(x)$. While the objective appears simple, in practice we do not have the prior knowledge of $p_t$ or $v_t$, and cannot directly compute the loss or its gradient estimator.

Let $x_1$ be a random variable distributed according to data distribution $q$. Lipman et al. [38] first notes that a probability path $p_t(x)$ can be constructed via a mixture of simpler *conditional paths* $p_t(x \mid x_1)$ whose vector field $u_t(x \mid x_1)$ can be easily computed. To construct $p_t(x)$, a conditional path is defined such that 1) $p_0(x \mid x_1) = p_0(x)$ and 2) $p_1(x \mid x_1) = \mathcal{N}(x \mid x_1, \sigma^2 I)$, a Gaussian distribution centered at $x_1$ with a sufficiently small $\sigma$ (typically $10^{-5}$). The marginal path is computed as $\int p_t(x \mid x_1) q(x_1) dx_1$, which closely approximates $q(x_1)$ at $t = 1$. With that, [38] presents the Conditional Flow Matching (CFM) objective, $\mathcal{L}_{CFM}(\theta) = \mathbb{E}_{t, q(x_1), p_t(x \mid x_1)} ||u_t(x \mid x_1) - v_t(x; \theta)||^2$.

It is proven that FM and CFM have identical gradients w.r.t. $\theta$. More importantly, one can easily draw samples from $p_t(x \mid x_1)$ and compute $u_t(x \mid x_1)$ to derive an unbiased gradient estimator.

The next question is *how to choose a conditional flow*. A flow defines *trajectories*, which dictates how each point moves between $p_0$ and $p_1$. Intuitively, a simpler trajectory (e.g., a straight line) can be learned faster and the IVP can be solved more accurately and efficiently. Lipman et al. [38] presents a conditional flow called *optimal transport (OT) path*, which has the form of $p_t(x \mid x_1) = \mathcal{N}(x \mid tx_1, (1 - (1 - \sigma_{min})t)^2 I)$ and $u_t(x \mid x_1) = (x_1 - (1 - \sigma_{\min})x) / (1 - (1 - \sigma_{\min})t)$. The flow is arguably simple because points move with a constant speed and direction. We adopt it for Voicebox.

Lipman et al. [38] also presents another flow that recovers the path of diffusion models [57], which is more complex than the OT path. We will present ablation studies comparing different paths (OT vs diffusion) and different objectives (CFM vs score-matching). Results show the superiority in performance and efficiency of CFM with OT path.

## 3.2 Problem formulation

Given a dataset of transcribed speech $(x, y)$ where $x$ and $y$ denote an audio sample and its transcript, respectively, the goal is to build a single model that can perform many text-guided speech generation tasks through in-context learning. We propose to train such a generative model on the *text-guided speech infilling task*, which predicts a segment of speech given its surrounding audio and the complete text transcript. Let $m$ be a binary temporal mask which is of the same length as $x$, and $x_{mis} = m \odot x$ and $x_{ctx} = (1 - m) \odot x$ be the complementary masked versions of $x$. The generative model learns $p(x_{mis} \mid y, x_{ctx})$. In other words, $y$ and $x_{ctx}$ are the context and $x_{mis}$ is the missing data.

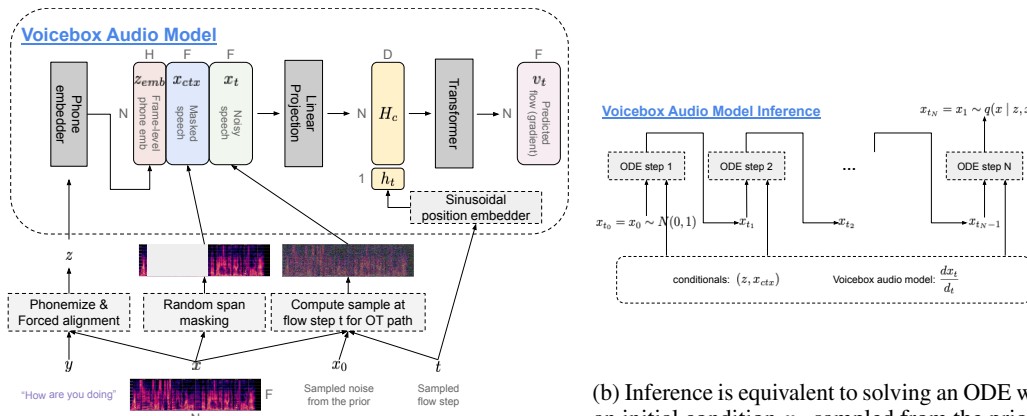

(a) Voicebox audio model. The lower half illustrates how inputs are created during training.

(b) Inference is equivalent to solving an ODE with an initial condition $x_0$ sampled from the prior, a derivative $\frac{dx_t}{dt}$ specified by the audio model, and conditional inputs $(z, x_{ctx})$.

Figure 2: Illustration of Voicebox training and inference.

## 3.3 Model and Training

Motivated by the need that some applications require fine-grained alignment control between speech and text, we decouple Voicebox into two components: an audio model and a duration model. Let $x = (x^1, x^2, \cdots, x^N)$ be an audio sample of $N$ frames, $y = (y^1, y^2, \cdots, y^M)$ be a text sequence of $M$ phones, and $l = (l^1, l^2, \cdots, l^M)$ be the per-phone duration where $l^j$ denotes how many audio frames $y^j$ correspond to and $\sum_{j=1}^{M} l^j = N$. We further define $z = \mathtt{rep}(y, l) = (z^1, z^2, \cdots, z^N)$ to be the frame-level phone transcript, which repeats each $y^j$ by $l^j$ times such that $z^i$ denotes the phone label of the audio frame $x^i$. For a pair of $(x, y)$, $l$ and $z$ can be estimated through *forced alignment* using a speech recognition model. The estimation of $q(x_{mis} \mid y, x_{ctx})$ is then broken down into the audio model $q(x_{mis} \mid z, x_{ctx})$ and the duration model $q(l_{mis} \mid y, l_{ctx})$, where $l_{mis}$ and $l_{ctx}$ denote $l$ masked by $m'$ and $1 - m'$, and $m'$ is downsampled from $m$ based on $l$, detailed in Appendix A.2.

**Audio Model:** Given a context $z$ and $x_{ctx}$ of length $N$, the distribution of $x_{mis}$ is highly stochastic especially when $x_{mis}$ has a large temporal span. Hence, we parameterize it with a CNF and train

it using the flow matching objective with the optimal transport path. Audio $x$ is represented as an 80-dimensional log Mel spectrogram ($x^i \in \mathbb{R}^{80}$) extracted at a 100Hz frame rate. The audio context $x_{ctx}^i = \mathbf{0}$ where $m^i = 1$ and $x_{ctx}^i = x^i$ where $m^i = 0$. For simpler conditioning, we model the conditional distribution $q(x \mid z, x_{ctx})$ of all frames $x$ instead of only masked frames $x_{mis}$. A neural network is used to parameterize the conditional vector field $v_t(x_t, x_{ctx}, z; \theta)$ that additionally takes $x_{ctx}$ and $z$ as input. Note that $x_t$ is a sample at flow step $t$ and $x = x_1$.

Given as input $x_{ctx} \in \mathbb{R}^{N \times F}$, $x_t \in \mathbb{R}^{N \times F}$, phone sequence $z \in [K]^N$ with $K$ denoting the number of phone classes, and a time step $t \in [0, 1]$, we employ a Transformer model to parameterize the vector field $v_t$. A lookup table, denoted as $L \in \mathbb{R}^{K \times H}$, is used to embed the phone sequence $z$, resulting in the embedded sequence $z_{emb} \in \mathbb{R}^{N \times H}$ where $z_{emb}^i = L(z^i)$ for $i \in 1, \ldots, N$. Subsequently, the three sequences ($x_t$, $x_{ctx}$, and $z_{emb}$) are concatenated frame-by-frame and projected by employing matrix $W_p \in \mathbb{R}^{(2F+H) \times D}$, thereby obtaining the sequence $H_c \in \mathbb{R}^{N \times D}$ where $D$ represents the embedding dimension of the Transformer model.

To embed the flow step, a sinusoidal positional encoding is applied to map $t \in [0, 1]$ to $h_t \in \mathbb{R}^D$. The sequence $\tilde{H}_c \in \mathbb{R}^{(N+1) \times D}$, which serves as the input to the Transformer model, is derived by concatenating $H_c$ with the vector $h_t$ along the time dimension. Given the Transformer output $v_t(x_t, x_{mis}, z; \theta) \in \mathbb{R}^{N \times F}$, which is the sub-sequence corresponding to $H_c$, the loss is computed as:

$$\mathcal{L}_{\text{audio-CFM}}(\theta) = \mathbb{E}_{t,m,q(x,z),p_0(x_0)} ||u_t(x_t \mid x) - v_t(x_t, x_{ctx}, z; \theta)||^2, \tag{1}$$

by reparameterization. During training, given an audio sample $x$ and a prior sample $x_0$, we have $x_t = (1 - (1 - \sigma_{\min})t)x_0 + tx$ and $u_t(x_t \mid x) = x - (1 - \sigma_{min})x_0$. This function computes the loss on all frames, including those that are not masked and would not be required during inference. To divert the model's focus to masked frames, we present a masked version of $\mathcal{L}_{\text{audio-CFM}}$:

$$\mathcal{L}_{\text{audio-CFM-m}}(\theta) = \mathbb{E}_{t,m,q(x,z),p_0(x_0)} ||m \odot (u_t(x_t \mid x) - v_t(x_t, x_{ctx}, z; \theta))||^2, \tag{2}$$

where the loss is only computed on masked frames. Appendix B.3 shows it leads to better results

**Duration model:** We consider two solutions. The first one closely follows the audio model. It models $q(l \mid y, l_{ctx})$ via a conditional vector field which swaps $(x, x_{ctx}, z)$ with $(l, l_{ctx}, y)$ and accordingly for the flow, where $l, l_{ctx} \in \mathbb{R}^{M \times 1}$ and $y \in [K]^M$. The masked version of the CFM loss is used for training. On the other hand, previous studies have shown that regression duration models can produce reasonable speech [51, 37]. Hence we consider a second solution that regresses the masked duration $l_{mis}$ given the context duration $l_{ctx}$ and phonetic transcript $y$. The same Transformer model is used, except that there are only two input sequences instead of three, and the time embedding is not used. The model is trained with an $L_1$ regression loss on masked phones:

$$\mathcal{L}_{\text{dur-regr-m}}(\theta) = \mathbb{E}_{m,q(l,y)} ||m' \odot (l_{mis} - g(l_{ctx}, y; \theta))||_1, \tag{3}$$

where $g$ denotes the regression-based duration model. This is similar to the duration model used in FastSpeech2 [51], but with additional duration context $l_{ctx}$ as input.

### 3.4 Inference

To sample from the the learned audio distribution $p_1(x \mid z, x_{ctx})$, a noise $x_0$ is first sampled from $p_0$, and then an ODE solver is used to evaluate $\phi_1(x_0)$ given $d\phi_t(x)/dt = v_t(\phi_t(x), x_{ctx}, z; \theta)$ and the initial condition $\phi_0(x_0) = x_0$. Intuitively, the ODE solver computes $\phi_1(x_0)$ by evaluating $v_t$ at multiple $t$ to approximate the integration from $t = 0$ to $t = 1$ given the initial condition $\phi_0(x_0) = x_0$. The number of function evaluation (NFE) is defined as how many times $d\phi_t(x_0)/dt$ is evaluated. A higher NFE often leads to a more accurate solution of $\phi_1(x_0)$ at the cost of longer run time. This provides great flexibility for users to decide the trade-off between speed and accuracy. Moreover, we find that empirically Voicebox can already generate very high quality speech with less than 10 NFEs, making it significantly faster compared to auto-regressive models.

### 3.5 Classifier-Free Guidance

Classifier guidance (CG) [14] is a technique used to trade off mode coverage and sample fidelity for diffusion models post training. It modifies the score estimate of a diffusion model to include the gradient of the log likelihood of an auxiliary classifier. Ho and Salimans [18] notes that CG approximates sampling from $p(x \mid c)p(c \mid x)^\alpha$ where $c$ is the conditioner, and this can be simulated

without a classifier by mixing the score estimate of a conditional model and an unconditional model. The unconditional model can be jointly trained by dropping the conditioner $c$ with some probability, and the same model provides score estimates for both $p(x)$ and $p(x \mid c)$.

We extend the idea of classifier free guidance (CFG) to flow-matching models. The conditioner $c$ is equivalent to $(z, x_{ctx})$ for audio models and $(y, l_{ctx})$ for duration models, which is dropped with $p_{\text{uncond}}$ during training. During inference, the modified vector field $\tilde{v}_t$ for the audio model becomes $\tilde{v}_t(w, x_{mis}, z; \theta) = (1 + \alpha) \cdot v_t(w, x_{ctx}, z; \theta) - \alpha \cdot v_t(w; \theta)$, where $\alpha$ is the strength of the guidance, and $v_t(w; \theta)$ is obtained by dropping $x_{ctx}$ and $z$. We use $\alpha$ and $\alpha_{dur}$ for the CFG strengths for the audio and the duration model, selected based on validation. Note that the computation is doubled for the same NFE when using CFG, because the model forward is called twice to compute $\tilde{v}_t$.

### 3.6 Applications

We demonstrate that Voicebox exhibits in-context learning abilities similar to LLMs by presenting a few examples of how to create context to perform tasks Voicebox was not explicitly trained on. Fig. A1 shows a detailed diagram of how inputs are formatted for each task.

**Zero-shot TTS & alignment-preserved style transfer:** Given a target text $\hat{y}$ and a transcribed reference audio $(x, y)$, zero-shot TTS aims to synthesize speech resembling the possibly unseen audio style of the reference. Voicebox performs the task by treating the reference audio and the target speech as one utterance where the target speech is masked. Let $l$ and $z$ be phone duration and frame-level transcript of $(x, y)$. The target duration $\hat{l}$ is sampled given the duration context $l$ and concatenated phone sequence $\texttt{cat}(y, \hat{y})$. The target speech $\hat{x}$ is then sampled given the context $x$ and concatenated frame-level phones $\texttt{cat}(z, \texttt{rep}(\hat{y}, \hat{l}))$.

Voicebox can also convert the audio style for speech $\bar{x}$ while preserving its alignment $\bar{z}$. This is useful for editing audio that is synchronized with other modalities such as video. Similar to zero-shot TTS, Voicebox can simply perform the task by sampling target speech $\hat{x}$ given the context $x$ and concatenated frame-level phones $\texttt{cat}(z, \bar{z})$

**Transient noise removal & content editing:** When recording speech, one might misspeak a few words or the recording my be interrupted by unexpected background noise. In these scenarios it is desired to just edit the problematic segment instead re-recording the speech. Voicebox can perform transient noise removal through re-generating the noise corrupted segment given the original frame-level transcript and the surrounding clean audio.

For content editing, Voicebox first samples duration for the new phones given the edited phone transcript and the duration of existing phones to create the edited frame-level phone transcript. Given the new frame-level phone transcript and the audio for existing frames, Voicebox then samples the audio for frames corresponding to the new phones.

**Diverse speech sampling & alignment-preserved style shuffling:** Voicebox can generate diverse speech samples by infilling the whole utterance. We first use the duration model to sample $\hat{l}$ given the phone transcript $\hat{y}$. We then use the audio model to sample $\hat{x}$ given $\hat{z} = \texttt{rep}(\hat{y}, \hat{l})$. Similar to style transfer, Voicebox can also shuffle the audio style while keeping the alignment by sampling $\hat{x}$ conditioning on the frame-level transcript $\bar{z}$ of the target speech clip $\bar{x}$.

## 4 Metrics

The common goal of audio-conditioned tasks is to produce *realistic* speech that is *coherent* with the context and has the *correct* textual content. For tasks not conditioned on audio context, it is desired to generate *diverse and realistic* samples with distribution similar to training data. Prior studies often adopt subjective metrics like mean opinion scores (MOS) [52] which are not comparable across papers, or quantitative metrics like mel cepstral distortion [35] that assume the output is deterministic given input, which is often not realistic [54]. In this paper, we advocate the following reproducible model-based perceptual metrics.

**Correctness and intelligibility:** We measure it by the word error rate (**WER**) of the synthesized speech's transcription with respect to the input text, which has been adopted in prior work [62]. Public automatic speech recognition (ASR) models are used for comparability. For English-only setups, we

follow [61] and use HuBERT-L [22] pre-trained on 60K hours of Librilight [26] and fine-tuned on 960 hours of Librispeech [43]. For multilingual setups we use the Whisper large-v2 model [49].

**Coherence:** This is measured by the similarity between the embedding of generated speech and that of the audio context, where different embedding models would reflect coherence of different attributes. VALL-E proposed to use WavLM-TDCNN speaker embedding model, which maps an audio clip to a fixed dimensional vector, to measure voice similarity. We consider the same model to compare with VALL-E. In particular, VALL-E reports similarity with respect to *resynthesized* audio context by its vocoder (Encodec-decoder), which we call **SIM-resyn (SIM-r)**. SIM-resyn is not comparable across models using different vocoders. Hence, we advocate for computing similarity against the original audio context, which we call **SIM-orig (SIM-o)**.

**Diversity and quality:** Fréchet Inception Score (FID) [17] is widely adopted for image generation evaluations, which captures the similarity between generated and real images at the distribution level in some feature space. A shorter distance implies the distributions are more similar and generally reflects *both* higher sample quality and diversity. We adapt the metric for speech by using self-supervised wav2vec 2.0 feature [2] and refer to it as Fréchet Speech Distance (**FSD**). We verify its effectiveness in Appendix C.1 along with alternative features.

As supplementary metrics, we include quality MOS (**QMOS**) for subjective audio quality evaluation, and similarity MOS (**SMOS**) for subjective audio similarity evaluation given pairs of prompt and system-generated audio clips. Both of which are in the scale of 1 to 5 with 5 being the best. The MOS instructions and standalone metrics for duration models can be found in Appendix C.

# 5 Experiment

**Data:** We train the English-only model on 60K hours ASR-transcribed English audiobooks and the multilingual model on 50K hours of multilingual audiobooks from six languages: English (En), French (Fr), German (De), Spanish (Es), Polish (Pl) and Portuguese (Pt). The two models are abbreviated as VB-En and VB-Multi. The Montreal Forced Aligner (MFA) [40] is used to phonemize and force align the transcript based on the MFA phone set. Word position postfixes are added. Audio is represented as a 80-dimensional log Mel spectrogram and a HiFi-GAN vocoder trained on the same 60K hours English speech is used to generate waveform. More details about phone representation, data transformation, and vocoder can be found in Appendix A1-A3.

**Model:** Transformer [59] with convolutional positional embedding [2] and symmetric bi-directional ALiBi self-attention bias [48] are used for both the audio and the duration model. ALiBi bias for the flow step $x_t$ is set to 0. More details in Appendix Appendix A.7. The audio model has 24 layers, 16 attention heads, 1024/4096 embedding/feed-forward network (FFN) dimension, 330M parameters. The duration model has 8 heads, 512/2048 embedding/FFN dimensions, with 8/10 layers for English/multilingual setup (28M/34M parameters in total).

**Training:** VB-En/VB-Multi audio models are trained for 500K/750K updates with an effective batch size of 240K frames. For training efficiency, audio length is capped at 1,600 frames and chunked randomly if length exceeds. Duration models are trained for 600K updates with an effective batch size of 60K frames. The Adam [33] optimizer is used with a peak learning rate of 1e-4, linearly warmed up for 5K steps and decays over the rest of training. The audio/duration sequence is masked with $p_{\text{drop}} = 0.3/0.2$, and otherwise a segment of $r\%$ sequence length is masked, where $r \sim \mathcal{U}[70, 100]/\mathcal{U}[10, 100]$. $p_{\text{uncond}}$ is set to 0.2 for audio/duration models.

**Inference:** The `torchdiffeq` [9] package is used, which implements both fixed and adaptive step ODE solvers. By default, the midpoint solver is used with a step size of 0.0625 (NFE=32). The regression duration model is used by default. Silence at both ends are trimmed to 0.1 second max.

**Baselines:** We consider three baselines: 1) VALL-E [61], SOTA for English zero-shot TTS trained on Librilight. 2) YourTTS [7], SOTA multilingual zero-shot TTS model trained on VCTK, LibriTTS, TTS-Portugese [8], and M-AILABS French. It is a flow-based model adapted from VITS [32] using a pre-trained speaker embedder. 3) A3T [3], SOTA for NAR speech editing and infilling trained with a regression loss on VCTK. We also consider Demucs [11], a SOTA speech enhancement model trained with regression and adversarial losses for denoising experiments.

## 5.1 Monolingual and cross-lingual zero-shot TTS

Table 2 presents the zero-shot TTS results of the English model VB-En. Following [61], the test set is constructed by selecting 4 to 10 second long samples from Librispeech test-clean. We consider *cross-sentence* prompting where a 3 second clip from another sample of the same speaker is used as audio context, and *continuation* prompting where the first 3 seconds of each utterance is used. Voicebox outperforms all baselines on all metrics in both cases. In particular, Voicebox transfers style much more effectively (+0.101/+0.108 SIM-r on cross-sentence/continuation) than VALL-E, and the gap is even bigger when compared against raw audio (+0.141 SIM-o on continuation). MOS studies also confirm the quality and similarity of Voicebox are subjectively better than YourTTS.

Table 2: English zero-shot TTS results on filtered LS test-clean. *obtained via personal communication.

| Model | WER | SIM-o | SIM-r | QMOS | SMOS |
|---|---|---|---|---|---|
| Ground truth | 2.2 | 0.754 | n/a | $3.98_{\pm 0.14}$ | $4.01_{\pm 0.09}$ |
| *cross-sentence* | | | | | |
| A3T | 63.3 | 0.046 | 0.146 | - | - |
| YourTTS | 7.7 | 0.337 | n/a | $3.27_{\pm 0.13}$ | $3.19_{\pm 0.14}$ |
| VALL-E | 5.9 | - | 0.580 | - | - |
| VB-En | 1.9 | 0.662 | 0.681 | $3.78_{\pm 0.10}$ | $3.71_{\pm 0.11}$ |
| *continuation* | | | | | |
| A3T | 18.7 | 0.058 | 0.144 | - | - |
| VALL-E | 3.8 | 0.452* | 0.508 | - | - |
| VB-En ($\alpha = 0.7$) | 2.0 | 0.593 | 0.616 | - | - |

Table 3: Transient noise removal where noise overlaps with 50% of the speech at a -10dB SNR.

| Model | WER | SIM-o | QMOS |
|---|---|---|---|
| Clean speech | 2.2 | 0.687 | $4.07_{\pm 0.15}$ |
| Noisy speech | 41.2 | 0.287 | $2.50_{\pm 0.15}$ |
| Demucs | 32.5 | 0.368 | $2.86_{\pm 0.17}$ |
| A3T | 11.5 | 0.148 | $3.10_{\pm 0.15}$ |
| VB-En ($\alpha = 0.7$) | 2.0 | 0.612 | $3.87_{\pm 0.17}$ |

Table 4 presents cross-lingual zero-shot TTS results, where the audio context and the target text are in different languages. Note that VB-Multi is *not* trained on any sample with multiple languages in an utterance spoken by the same speaker. The test set is constructed using filtered MLS test split described in Appendix A.4. For each target text, we sample one 3-second long audio context from each language, which creates 36 language transfer directions in total. Voicebox yields better performance than YourTTS everywhere. Specifically, on En/Fr/Pt which YourTTS supports, Voicebox obtains 3.1%/5.9%/8.1% lower WERs and 0.136/0.141/0.160 higher similarity averaged across audio context in six languages. Addition studies on prompt lengths are presented in Appendix B.2

Table 4: Multilingual zero-shot TTS results on filtered MLS test sets. GT/YT/VB-Multi refers to ground truth/YourTTS/multilingual Voicebox. "Ref" column shows the audio context language.

| | Ref | De WER | De SIM-o | En WER | En SIM-o | Es WER | Es SIM-o | Fr WER | Fr SIM-o | Pl WER | Pl SIM-o | Pt WER | Pt SIM-o |
|---|---|---|---|---|---|---|---|---|---|---|---|---|---|
| GT | - | 5.9 | 0.725 | 5.0 | 0.636 | 4.1 | 0.729 | 5.2 | 0.714 | 4.9 | 0.743 | 5.8 | 0.725 |
| | De | n/a | n/a | 7.3 | 0.373 | n/a | n/a | 11.3 | 0.361 | n/a | n/a | 13.7 | 0.263 |
| | En | n/a | n/a | 7.0 | 0.403 | n/a | n/a | 11.4 | 0.298 | n/a | n/a | 14.1 | 0.234 |
| | Es | n/a | n/a | 7.6 | 0.327 | n/a | n/a | 11.6 | 0.316 | n/a | n/a | 13.5 | 0.256 |
| YT | Fr | n/a | n/a | 7.6 | 0.363 | n/a | n/a | 10.7 | 0.459 | n/a | n/a | 13.1 | 0.299 |
| | Pl | n/a | n/a | 7.8 | 0.349 | n/a | n/a | 11.8 | 0.370 | n/a | n/a | 15.1 | 0.308 |
| | Pt | n/a | n/a | 7.6 | 0.322 | n/a | n/a | 11.8 | 0.297 | n/a | n/a | 13.6 | 0.436 |
| | AVG | n/a | n/a | 7.5 | 0.356 | n/a | n/a | 11.4 | 0.350 | n/a | n/a | 13.9 | 0.299 |
| | De | 4.8 | 0.632 | 4.8 | 0.522 | 3.6 | 0.442 | 5.3 | 0.489 | 5.5 | 0.449 | 5.4 | 0.420 |
| | En | 5.9 | 0.435 | 4.2 | 0.535 | 4.1 | 0.423 | 6.8 | 0.423 | 8.3 | 0.402 | 7.6 | 0.385 |
| VB-Multi | Es | 4.9 | 0.460 | 4.3 | 0.479 | 3.6 | 0.613 | 5.3 | 0.473 | 5.2 | 0.436 | 5.4 | 0.435 |
| ($\alpha = 1.0$) | Fr | 4.9 | 0.476 | 4.3 | 0.485 | 3.7 | 0.479 | 5.1 | 0.602 | 4.8 | 0.408 | 5.4 | 0.418 |
| | Pl | 4.7 | 0.491 | 3.8 | 0.503 | 3.5 | 0.528 | 5.1 | 0.503 | 4.0 | 0.641 | 4.9 | 0.476 |
| | Pt | 4.9 | 0.422 | 4.6 | 0.426 | 3.7 | 0.476 | 5.5 | 0.453 | 4.8 | 0.406 | 5.2 | 0.620 |
| | AVG | 5.0 | 0.486 | 4.4 | 0.492 | 3.7 | 0.494 | 5.5 | 0.491 | 5.5 | 0.457 | 5.7 | 0.459 |

## 5.2 Transient noise removal

We construct a noisy test set by mixing the filtered Librispeech test-clean from Section 5.1 with non-speech noise such that it overlaps with 50% of the duration at a -10dB signal-to-noise ratio. Additional conditions can be found in Appendix B.4. Table 3 presents the results comparing Voicebox with A3T and Demucs. It should be noted that A3T and Voicebox utilize transcript and location of the noise while Demucs does not. Compared to the baselines, Voicebox generates samples best on all metrics. A3T is better than Demucs on intelligibilty and quality, but the infilled speech is not coherent because it is only trained on VCTK.

## 5.3 Diverse speech sampling and application to ASR data generation

Table 5 compares the ability to generate diverse samples for Librispeech test-other text. We consider English Voicebox (VB-En) with regression (regr) or flow-matching (FM) duration models. VITS-VCTK additionally conditions on a speaker ID, which we randomly sample for each sentence. YourTTS conditions on text and a reference audio, which we draw from the LS train splits. Qualitatively, A3T generates the same robotic voice and VITS-LJ generates high quality but from a single voice, hence both yield high FSD (bad quality or diversity) but VITS-LJ has a low WER. VITS-VCTK improves the voice diversity and FSD and YourTTS further advances it as it is trained on more speakers. Voicebox models (with different duration samplers) outperform the baseline on FSD by large margins, showing Voicebox's ability to produce realistic and diverse samples whose distribution is close to the training data. Among them, the FM duration model creates more varying speaking styles compared to the regression one which ASR may struggle more to recognize.

Table 5: Diverse speech generation from LS test-other text.

| Model | WER | FSD |
|---|---|---|
| Ground truth | 4.3 | 171.1 |
| *require additional input* | | |
| VITS-VCTK | 10.6 | 306.6 |
| YourTTS (ref=LS train) | 9.0 | 277.9 |
| *text-only* | | |
| A3T | 37.9 | 373.0 |
| VITS-LJ | 5.6 | 344.2 |
| VB-En ($\alpha = 0$, dur=regr) | 3.1 | 155.7 |
| VB-En ($\alpha = 0$, dur=FM, $\alpha_{dur} = 0$) | 5.6 | 159.8 |

Table 6: Performance of ASR models trained on real or synthetic speech, tested on *real* speech and decoded with or without a 4-gram language model.

| | WER on real data | | | |
|---|---|---|---|---|
| | No LM | | 4-gram LM | |
| ASR training data | test-c | test-o | test-c | test-o |
| Real audio (100hr) | 9.0 | 21.5 | 6.1 | 16.2 |
| Real audio (960hr) | 2.6 | 6.3 | 2.2 | 5.0 |
| VITS-LJ | 58.0 | 81.2 | 51.6 | 78.1 |
| VITS-VCTK | 33.8 | 55.5 | 30.2 | 53.1 |
| YourTTS (ref=LS train) | 25.0 | 54.6 | 20.4 | 51.2 |
| VB-En ($\alpha = 0$, dur=regr) | 7.1 | 17.6 | 6.5 | 14.6 |
| VB-En ($\alpha = 0$, dur=FM, $\alpha_{dur} = 0$) | 3.1 | 8.3 | 2.6 | 6.7 |

We next train an ASR model using *only synthetic speech* and evaluate it on *real speech*, which has not been successful before because synthetic data were not realistic and representative enough. Table 6 compares real and synthetic data from Voicebox and three baseline models. Each TTS model generates one sample per text from the Librispeech training set, resulting in 281K utterances per system. For real data, we consider train-960 and train-clean-100. Details about the ASR model and training are in Appendix A.5. The results are highly correlated with the FSD scores for synthetic data. In particular, the ASR model trained on Voicebox data with FM duration model reduces WERs by over 85% compared to baselines, and only lags behind real data by 0.4% and 1.7% absolute.

## 5.4 Inference efficiency versus performance

We examine the trade-off between the metrics of interest (WER, SIM, FSD) for different settings of guidance strength ($\alpha$) and NFE specified by the user. Fig. 3a shows the Voicebox inference time to generate an audio sample of 10 seconds (including vocoding and predicting duration) as NFE varies and compares that to VALL-E.[3] For NFE=2 without CFG, Voicebox takes about 0.31 seconds, about 20 times faster than VALL-E. At NFE=64, Voicebox is only 4% slower than VALL-E.

Next, we study the cross-sentence setup of Section 5.1 to analyze the impact on WER and SIM-r. We find that for all settings Voicebox has better WER than VALL-E. WER remains stable with mean of 2.0 and variance of 0.005. WER plot can be found in Appendix B.5. As shown in Fig. 3b, in the case of SIM-r, lower classifier guidance strength values ($\alpha = 0$ or $0.3$) produce higher speaker similarity when operating in a lower NFE regime ($\leq 4$). However, starting from NFE=8, a higher classifier guidance strength improves speaker similarity. Finally, in Fig. 3c we examine FSD by generating samples for Librispeech test-other text. We find that lower classifier guidance strength produces lower FSD scores and more diverse samples. Increasing the NFE for each setting improves FSD.

## 5.5 Ablation on generative modeling approaches

We compare three generative modeling approaches: the proposed flow-matching with the OT path (FM w/ OT), flow-matching with the variance preserving (VP) diffusion path (FM w/ diff), and score-

---

[3]Re-implemented and confirmed with the authors that our re-implementation is faster (6.2 vs 10 seconds).

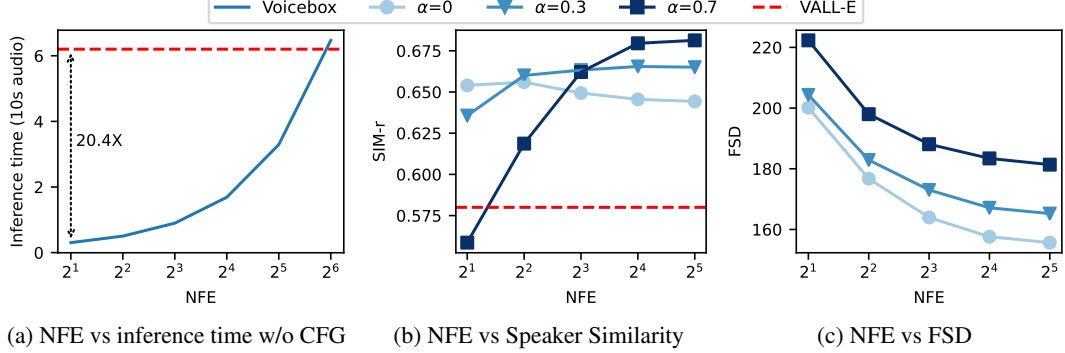

(a) NFE vs inference time w/o CFG  (b) NFE vs Speaker Similarity  (c) NFE vs FSD

Figure 3: Trade-off between NFE and different metrics. Inference time will be doubled with CFG.

Table 7: Comparing different objectives on training efficiency. 32 NFEs are used for inference. Each model is evaluated on the monolingual zero-shot TTS task.

| Method | upd=50K | | upd=100K | | upd=150K | |
|---|---|---|---|---|---|---|
| | WER | Sim-o | WER | Sim-o | WER | Sim-o |
| FM w/ OT (proposed) | **2.5** | **0.424** | **2.2** | **0.487** | **2.1** | **0.508** |
| FM w/ diff | 76.0 | 0.066 | 3.1 | 0.344 | 2.6 | 0.478 |
| SM w/ diff | 73.3 | 0.062 | 17.4 | 0.176 | 5.1 | 0.349 |

Table 8: Comparing different objectives on inference efficiency. All models are trained for 150K updates. Each model is evaluated on the monolingual zero-shot TTS task.

| Method | NFE=4 | | NFE=8 | | NFE=16 | | NFE=32 | |
|---|---|---|---|---|---|---|---|---|
| | WER | Sim-o | WER | Sim-o | WER | Sim-o | WER | Sim-o |
| FM w/ OT (proposed) | **2.4** | **0.410** | **2.2** | **0.481** | **2.2** | **0.503** | **2.1** | **0.508** |
| FM w/ diff | 11.5 | 0.171 | 3.0 | 0.359 | 2.7 | 0.447 | 2.6 | 0.478 |
| SM w/ diff | 94.5 | 0.054 | 42.3 | 0.076 | 11.5 | 0.218 | 5.1 | 0.349 |

matching with the VP diffusion path (SM w/ diff). A reduced setup described in B.3 is adopted, with a lowered learning rate (1e-4) and the loss in Eq. (1) to ensure convergence for all three objectives.

We vary the number of training and inference steps, and evaluate models on the zero-shot TTS task (Section 5.1). Results in Table 7 shows that FM w/ OT trains significantly faster than the other two objectives, achieving the best performance with 100K training steps, and even outperforms SM w/ diff using only 50K updates. Results in Table 8 shows superior inference efficiency of FM w/ OT, which can produce good results with just 8 NFEs, while FM w/ diff requires at least 8 NFEs and SM w/ diff requires over 32 NFEs. Complete results are in Table B5

## 6 Conclusion

This paper presents Voicebox, the most versatile generative model for speech. By learning to solve a text-guided speech infilling task on large scale multilingual datasets with a power model and training objective Voicebox demonstrates impressive task generalization capabilities. Voicebox achieves state-of-the-art performance on mono and cross-lingual zero-shot TTS, speech inpainting, and diverse speech sampling, and can generate speech 20 times faster than the best autoregressive models.

With high fidelity speech generation models like Voicebox, it brings the potential of misuse and unintended harm. To mitigate the risk, we also detail in Appendix B.1 that a highly effective classifier can be built to distinguish between authentic and synthetic speech. Voicebox is now trained only on read speech from audiobooks in six languages, and cannot transfer one attributes (e.g., emotion) from a reference while transferring another attribute (e.g., voice) from another reference. Due to space limit, we expand our discussion of limitation and broader impact in Appendix D. For future work, we would continue scaling the model and the data to include more languages and diverse types of speech such as conversations, and explore disentangled prompting for different attributes.

## Acknowledgment

The authors would like to thank Kristin Lauter and Joelle Pineau for supporting the project, thank Ricky Chen, Yaron Lipman, Alexandre Defossez, Gabriel Synnaeve for the technical discussion, thank Jade Copet and Gabriel Synnaeve for the compute support, thank Eleonora Presani and Jackie Pan for discussing the responsible AI studies, thank William Ngan, Somya Jain, Lydia Baillergeau, Dana Beaty, Chantal Mora, Daniel Duncan, Gopika Jhala, Steph Miles, Josh Terry, Valeryia Aranovich, Ashton Evans, Aly Gill, Andrea Mileskiewicz, Emily Richards, and Aaron Vasquez for developing the visual assets and the website, thank Alyssa Newcomb and Oliver Libaw for developing the posts, thank Peter Gray, Natalie Hereth, Shauna Kelleher, Ashley Gabriel, Seine Kim, Ana Paula Kirschner Moffarej and Aiman Farooq for coordinating the launch, thank Harrison Rudolph, Mallika Malhotra, Carolyn Krol, Lauren Cohen and Mo Metanat for the reviews, and thank Alexandra Gualdino, Ana Paula Kirschner Mofarrej, Benjamin Muller, Chloe Rolland, Daniella Kalfa, Darcie Da Silva, Gabriel Synnaeve, Hunter Goldman, Juan Pino, Karen Ulrich, Kris Sekula, Manuel Ribeiro, Marina Zannoli, Mary Williamson, Rashel Moritz, Stephanie Castillo, Tu Anh Nguyen, Vlad Sobal, Volker Seeker, and anonymous volunteers for sharing speech samples for the demo.

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

# A  Additional Details of Experiment Setup

## A.1  Vocoder

We adapt the HiFi-GAN V1 configuration to generate 16kHz audio from 80 dimensional log Mel spectral features sampled at 100Hz. To compute the log Mel spectrogram, we use a 1024-point short time Fourier transform with a 640-sample (40ms) analysis window, 160-sample (10ms) shift, and the Hann windowing function to compute the amplitude spectrogram, and then apply an 80 dimension Mel filter with a cutoff frequency at 8kHz. The original HiFi-GAN V1 has four transposed convolution blocks for upsampling. The upsampling factors are $[8, 8, 2, 2]$ and the corresponding kernel sizes are $[16, 16, 4, 4]$. Here we only need a total upsampling factor of 160 instead of 256, and we adjust the upsampling factors to $[5, 4, 4, 2]$ and kernel sizes to $[11, 8, 8, 4]$ accordingly. The other parameters are identical to the HiFi-GAN V1 configuration. Total number of parameters is 13M. We train the adapted HiFi-GAN on the 60K hours of English audiobook data for 1.5M updates on 8 GPUs, which takes 7.5 days.

## A.2  Phone representation

**Ghost silence**   The frame-level phonetic transcript used for training is obtained through force-aligning speech and phonetic transcript. In particular, a forced aligner may align some frames to a special phone "SIL" for non-speech frames (silence or noise). For most forced aligners, only frames between words and frames at the beginning and at the end of an utterance can be aligned to SIL.

During inference, we are only given the text transcript, which does not tell us where we should insert silence to. Hence, it is desired to have the duration model not only predict the duration for each phone (SIL included), but also predict the *existence* of SIL at eligible locations (between words and at the two ends of the utterance). To tackle it, we introduce *ghost silence* to our phonetic transcript, which are silences in between words with duration of zero frames.

To give an example, suppose the transcript contains three words: "Hey what's up" with pronunciation "{Hey:[A,B], what's:[C], up:[D,E,F]}", and the frame-level phonetic transcript $z$ obtained through forced alignment is $z = $ (SIL A B B SIL C D D D E E F SIL SIL). The phonetic transcripts becomes $y = $ (SIL A B SIL C SIL D E F SIL), where the ghost silence is highlighted in green. The corresponding duration would be $l = (1, 1, 2, 1, 1, 0, 3, 2, 1, 2)$. A ghost silence is inserted between what's and up during training, and the duration model should predict the duration of it as zero to indicate that there should not be a pause between the two words.

**Word-position-dependent phone**   The possible absence of silence between words in the frame-level phone transcript can make it hard for the audio model to identify word boundaries. To help the audio model identify the word boundary which is important when reading a sentence, we introduce word-position-dependent phones which are commonly used in Hidden Markov Model based acoustic models for speech recognition [47]. This adds a postfix to each phone in the transcript to denote where it is in the corresponding word. There are four postfixes: _B for beginning, _E for end, _I for intermediate, and _S for singleton. The above example becomes "{Hey:[A_B,B_E], what's:[C_S], up:[D_B,E_I,F_E]}" with frame-level phonetic transcript $z = $ (SIL A_B B_E B_E SIL C_S D_B D_B D_B E_I E_I F_E SIL SIL).

**Phone-level mask**   In terms of masking, given duration $l$, the relationship of phone-level mask $m'$ and frame-level mask $m$ can be written as $m = \texttt{rep}(m', l)$. For the applications where a duration model is involved (zero-shot TTS, content editing, diverse speech sampling), the frame-level mask $m$ is extended such that no phone is partially masked. In other words, all the frames corresponding to a phone is either entirely masked or entirely unmasked. During training, we mask a contiguous chunk of audio, infilling of which is a more challenging task compared to infilling multiple smaller segments. All frames that are aligned to a phone are either entirely masked or unmasked. Note that masking all frames for a phone is not a necessity but was chosen due to ease of implementation.

## A.3  Data transformation

The Mel spectrogram is normalized with the global mean (-5.8843) and standard deviation (2.2615) to stabilize training. The statistics are estimated on 30k random training samples from the 1K hours of

English audio. Input and output duration are dequantized ($x \sim \mathcal{U}[x - 0.5, x + 0.5]$) and transformed with $\log(1 + x)$ following [51]. Prediction of duration is quantized and clipped such that the minimal duration is greater than or equal to zero.

## A.4 Cross-lingual zero-shot TTS test data filtering

We create a test set for each language by selecting samples from the MLS test split which have Whisper transcription WER lower than 20% (or 30% for Polish and Portugueses test splits which contains less than 1K samples), because we found MLS test set contains many examples with incomplete transcriptions missing a large portion of the utterance. In addition, a small amount of utterances were excluded due to MFA alignment failure. Table A1 lists the number of samples remained for each language.

Table A1: Number of MLS test samples after filtering.

| Language | #samples before filtering | #samples after filtering |
|---|---|---|
| English | 3769 | 3535 |
| Spanish | 2385 | 2323 |
| German | 3394 | 3183 |
| French | 2426 | 2284 |
| Polish | 520 | 508 |
| Portuguese | 871 | 838 |

## A.5 Setup for training ASR models with synthetic speech

To train an ASR model in Section 5.3, we extract 80-dimensional log Mel features with a 25ms window and a 10ms frame shift, and then apply global mean-variance normalization. The ASR model is an RNN-T with a Conformer-based encoder [16]. The conformer applies time scale reduction to the input features with stride 6, embeds them into 512-dimensional vectors, passes these vectors through a 20-layer conformer which has 8 attention heads and 2048-dimensional fully-connected layers. The conformer output is further mapped to 1024 dimensions through a linear layer followed by layer normalization before being passed to the joiner. The predictor of the network first embeds wordpiece units into 512 dimensional embeddings, applies layer normalization, a 512-dimensional LSTM, a dropout layer and a linear layer that maps the LSTM output to 1024 dimensions. The joiner adds the encoder and predictor outputs, applies tanh non-linearity and uses a linear layer that maps the 512-dimensional joiner input into wordpiece units. There are 4096 wordpiece units estimated from the LibriSpeech 960hr training text.

We apply SpecAugment [44] in all ASR runs. The models are trained using PyTorch [45] with Adam [33] optimizer for 120 epochs unless otherwise noted. The learning rate follows a tri-stage schedule with a maximum of 0.001. We applied gradient clipping at 10 and a weight decay parameter of 0.1. For the 960hr setting, we used a variable batch size capped at 1K utterances or 30K frames, whichever is smaller. This corresponds to about 45K update steps for 120 epochs. For the 100hr setting, we set the maximum learning rate to 0.0001 and used smaller batch size (capped at 200 utterances or 5K frames). In this case, 120 epochs corresponded to about 120K updates. For decoding, we used n-best decoding with a beam-size of 15, and evaluated the WER on the 1-best path.

## A.6 How to format input for different task

Fig. A1 shows how input are formatted to perform diverse speech sampling, content editing, and zero-shot TTS using Voicebox.

## A.7 Bi-directional ALiBi Bias

We use a symmetric variant bi-directional variant of ALiBi bias where any query $Q_i$ and key $K_j$ with $|i - j| = N$ use the same representations [4]. Furthermore, for any query $Q_i$ the bias corresponding

---

[4]Our implementation is similar to symmetric option.

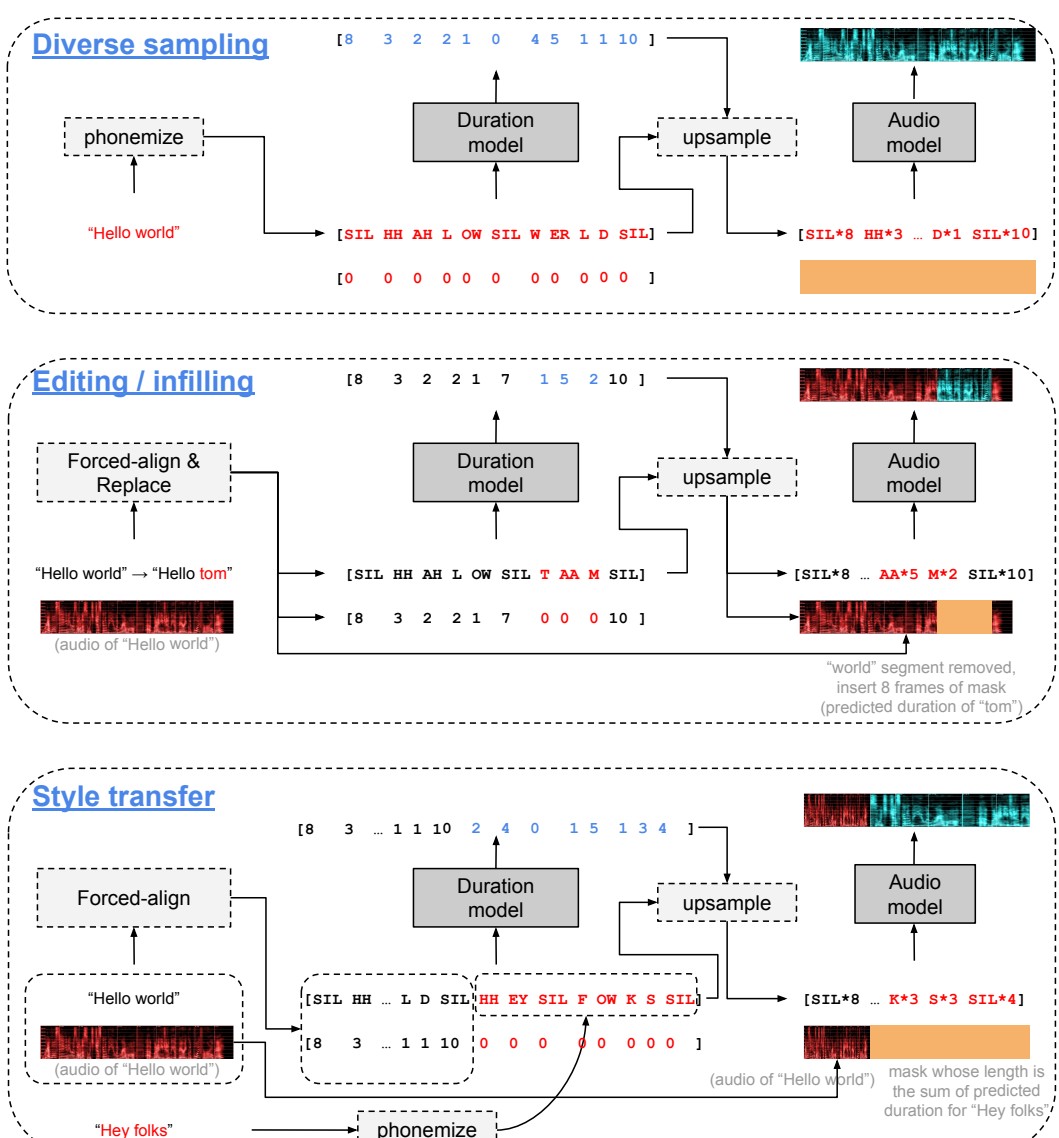

Figure A1: Detailed diagrams of diverse speech sampling, content editing, and style transfer. Text in red and blocks in orange at the input of a model denote segments to be predicted. Numbers in blue and spectrogram in cyan at the model output denote predicted duration and spectrogram.

to the flow step $x_t$ is set to 0. Similarly the bias from the flow-step $x_t$ to any other query 0. In our experiments, we find ALiBi Bias to improve convergence and extrapolation to longer sequences.

# B    Additional Experiments

## B.1    Detecting generated speech

We recognize the potential risks of a model capable of generating speech in the style of arbitrary users. In an effort to diminish these risks we show that a binary classification model is able to consistently distinguish between real world speech and that which is generated from our model.

Inspired by [29], we train a convolutional binary classification model to distinguish between real and generated speech. The model consists of 6 blocks with hidden dimension sizes: [64, 128, 256, 256, 512, 512]. Each block contains a (3 x 1) convolution along the time axis, a (1 x 3) convolution along the frequency axis, followed by a ReLU activation and batch normalization. After each block that

increases the hidden dimension size we also apply max pooling with a stride of 2 across both the time and frequency dimensions. Finally, global max pooling is applied and a linear layer projects to a single value that is fed into a binary cross entropy loss. At inference time we create a sliding window with hop length equal to 250ms and run each chunk of audio through the classifier and average the outputs.

The model is tested on the dev-clean split of Librispeech. We then take a 100 hour subset of the 60K hour-English data and set aside 2,703 random utterances (to match the size of dev-clean) which is used as a validation split. The remaining utterances from the 100 hours subset are used as the ground truth utterances for training. For each split we synthesize audio, conditioned on each utterance of the split by masking out frames in the spectrogram corresponding to 90%, 50%, and 30% of the phonemes of the utterance. All samples are generated using classifier-free guidance with $w = 0.7$, midpoint ODE solver (step size 0.0625 / NFE=64), and the regression duration model.

We consider two detection tasks. The first one is to distinguish between original audio and Voicebox-generated audio. The second one is to distinguish resynthesized audio and Voicebox-generated audio. The resynthesized audio is created by extracting the Mel Spectrogram from original audio and then vocoding it with the HiFi-GAN vocoder.

Table B2 presents the results for each setting. The model can trivially distinguish original audio from Voicebox-generated audio. This results from the fact that a model can also trivially distinguish original audio from resynthesized audio, most likely by recognizing artifacts produced by the vocoder. The task of differentiating Voicebox-generated audio from resynthesized audio is much harder. When 90% of the audio is masked, the model is able to reliably classify the audio as Voicebox-generated. In lower masking regimes this decreases a bit, but this is likely due to a naive inference method of averaging the outputs of all sliding windows. Since the majority of windows are non-synthetic, this leads to mis-classifications.

Table B2: Synthetic speech detection metrics

| % Mask | Accuracy | Precision | Recall |
|--------|----------|-----------|--------|
| *Original audio vs Voicebox-generated audio* | | | |
| 30% | 1.000 | 1.000 | 1.000 |
| 50% | 1.000 | 1.000 | 1.000 |
| 90% | 1.000 | 1.000 | 1.000 |
| *Resynthesized audio vs Voicebox-generated audio* | | | |
| 30% | 0.704 | 0.714 | 0.680 |
| 50% | 0.809 | 0.796 | 0.831 |
| 90% | 0.907 | 0.881 | 0.942 |

### B.2    How context length affects monolingual and cross-lingual zero-shot TTS

**Monolingual:** For in-context zero-shot TTS in Section 5.1, we used $3.0$ seconds of prompt audio. Here we examine how WER / SIM-r vary with different amounts of prompt audio using duration from regression duration model for the target text. If the desired prompt is longer than the available audio, the shorter audio is used as the prompt. Results are shown in Figure B2. As expected, WER mildly decreases and SIM-r grows quickly flattens with longer audio prompts. Comparing against VALL-E, Voicebox is more efficient at leveraging an audio prompt, achieving the same speaker similarity as VALL-E with roughly two thirds the input audio.

**Cross-lingual:** Here we examine the effect of increasing the prompt length for the case of cross-lingual zero-shot TTS. As described in 5.1, this setting has a total 36 language transfer directions for each pair of source and target language. For each target text in a given transfer setting, we examine how WER / SIM-o[5] vary as the prompt length increases. Similarly, the regression duration model is used for the target text. Fig. B3 and Fig. B4 plot the SIM-o (speaker similarity) and WER trends respectively. When concatenating the prompt to the target for MLS, we find that the samples are quite a bit longer than what the model was trained on (16s max length), because MLS test set samples are in average 15 seconds long. To alleviate this out of domain issue and focus the study

---

[5]Same trend is observed with SIM-r. We present SIM-o to be consistent with Table 4

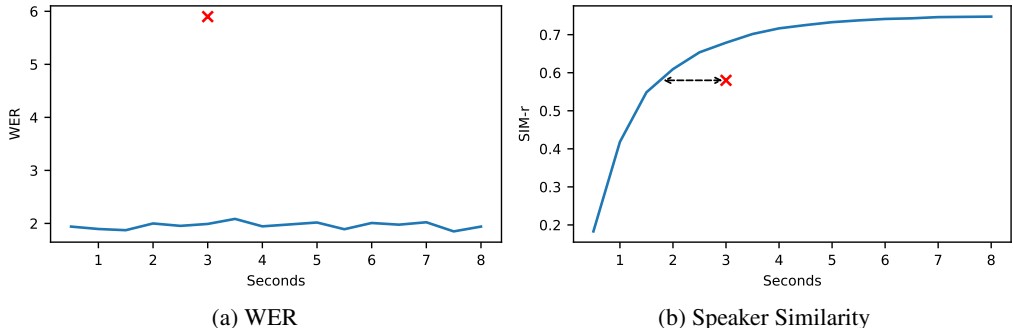

(a) WER

(b) Speaker Similarity

Figure B2: WER and SIM-r as a function of prompt audio time in seconds for the Zero-shot TTS task 5.1. Audio is generated using classifier-free guidance strength ($\alpha$) of 0.7 and midpoint ODE solver with a NFE of 32. The blue line is for Voicebox and the red star is VALLE at 3 seconds. The speaker similarity (SIM-r) remains same for longer prompts (up to 10s).

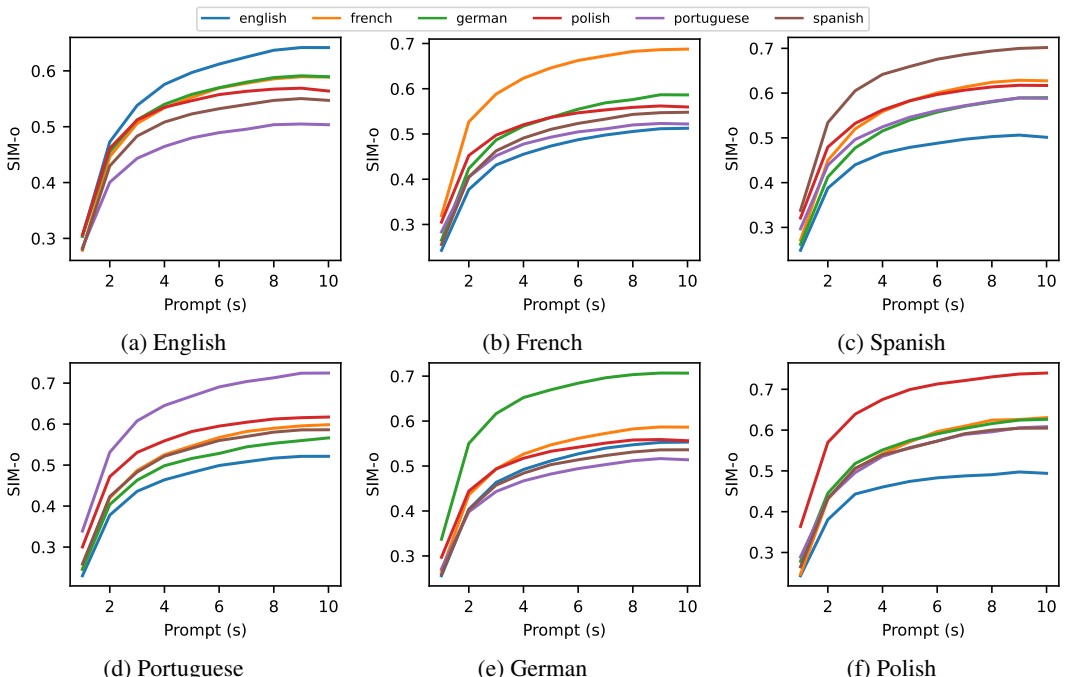

(a) English

(b) French

(c) Spanish

(d) Portuguese

(e) German

(f) Polish

Figure B3: Each subplot considers one of the six target language and shows SIM-o (speaker similarity) as a function of prompt audio duration in seconds for cross-lingual style transfer from different source language. We set the classifier-free guidance strength ($\alpha$) to 1.0 and use midpoint ODE solver with a NFE of 32.

on varying the prompt length, we truncate the target sequences to 4 seconds (at word boundaries). We notice that WERs are higher compared to Table 4, likely because the ASR model struggles with incomplete sentences. Each subplot contains the trend for one of the target languages from all six source languages.

The speaker similarity consistently improves as the prompt length is increased, similar to the monolingual setting. In contrast, we find that WER increases as we increase the prompt length for most directions. The WER increases much more for En → non-En directions. We hypothesize that this is due to training data imbalance across languages, where English accounts for over 90% of the multilingual training data. Hence, when transferring from English, the model is more likely to assume that the whole sentence is in English as the prompt length increases and produce incorrect

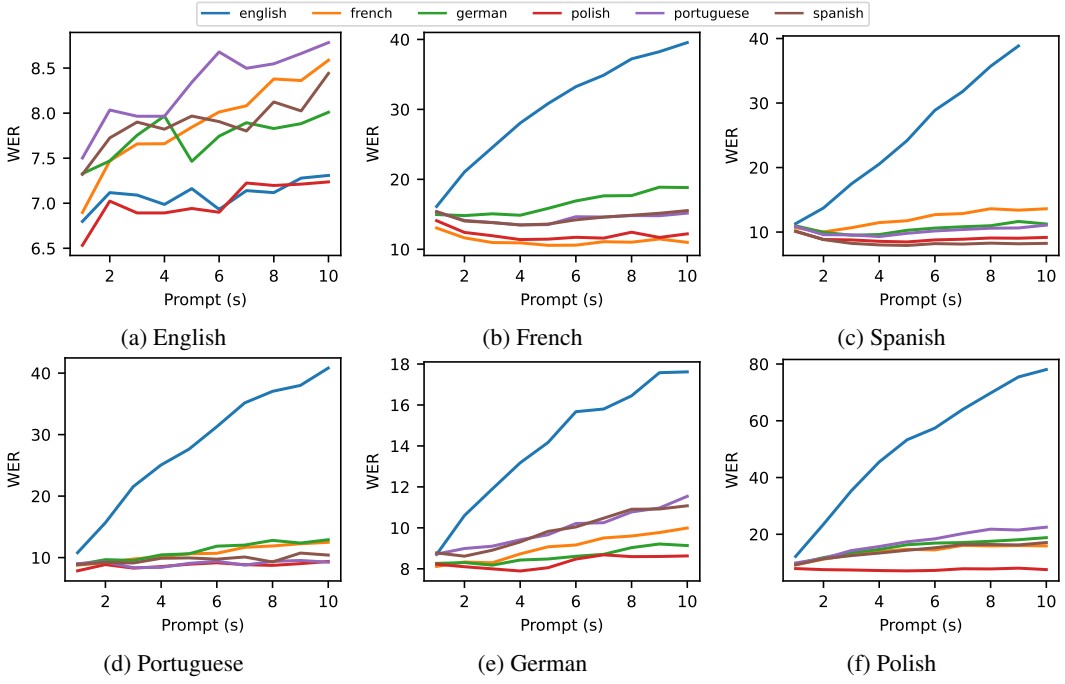

Figure B4: Each subplot considers one of the six target language and shows WER as a function of prompt audio duration in seconds for cross-lingual style transfer from different source language. We find WER remain reasonably low for all cases except for "English" to "X" style transfer. We set the classifier-free guidance strength ($\alpha$) to $1.0$ and use midpoint ODE solver with a NFE of $32$.

pronunciation for the non-English target. Note that during the training phase, the model was only exposed to audio samples and phonemes originating from a single language.

## B.3 Comparing audio model training objectives

While A3T is considered the regression-based speech infilling baseline, it is trained on a smaller dataset and uses a smaller model compared to Voicebox. Here we present a controlled study comparing the flow-matching and regression objectives, as well as the effectiveness of masked loss.

We consider a reduced setup for this ablation to save the compute. All models were trained on an English audiobook dataset with 1K hours of speech using a smaller model configuration (12 layers, 1024-dimensional Transformer embedding, 2048-dimensional feed-forward layer, 8 attention heads) for 150k steps with an effective batch size of 120k frames. These models are evaluated on the cross-sentence zero-shot TTS setup (Section 5.1) and diverse speech sampling (Section 5.3).

Results in Table B3 show that while regression audio models produce comparable WER, the audio similarity and diversity are significantly worse. Subjective listening also reveals that the audio quality and audio similarity are much worse. On the other hand, masked loss improves audio similarity and diversity while having little impact on intelligibility.

Table B3: Comparison of flow-matching and regression models, trained with loss computed on all frames or only masked frames. Results of the proposed objective is boldfaced.

| Method | Loss | Zero-Shot TTS (*cross-sentence*) | | Diverse sampling | |
| | | WER | SIM-r | WER | FSD |
| --- | --- | --- | --- | --- | --- |
| Flow Matching | Masked | **2.1** | **0.597** | **3.1** | **242.5** |
| Flow Matching | All | 2.0 | 0.528 | 3.1 | 243.1 |
| Regression | Masked | 2.0 | 0.520 | 2.9 | 278.8 |
| Regression | All | 2.0 | 0.512 | 2.9 | 282.8 |

## B.4 Effectiveness on data scaling

We create four subsets of the 60K hour English data (0.1%, 1%, 10%, 100% in duration). In particular, the $x\%$ subset would contain roughly $x\%$ of the speakers from the original set. We train one model on each subset with a reduced setup described in Appendix B.3 and evaluate them on zero-shot TTS (cross-sentence) and diverse sampling. Results show that scaling data constantly improves the zero-shot TTS performance (WER and SIM-r) as well as WER on diverse sampling. For FSD it shows regression when scaling from 6K hour to 60K hour, but this could result from the the reference distribution is computed from the 1K hour English audiobook data that has less diverse samples.

Table B4: Experiments on the effect of scaling training data.

| Train data (hr) | Zero-Shot TTS | | Diverse sampling | |
|---|---|---|---|---|
| | WER | SIM-r | WER | FSD |
| 60 | 2.30 | 0.151 | 3.48 | 280.48 |
| 600 | 2.11 | 0.417 | 3.19 | 205.39 |
| 6,000 | 2.08 | 0.573 | 2.96 | 195.52 |
| 60,000 | 2.05 | 0.645 | 2.95 | 214.38 |

## B.5 Complete results on comparing generative modeling approaches

Table B5 presents the full results of Section 5.5 on all combinations of training steps, inference steps with results on both monolingual zero-shot TTS (Section 5.1) and diverse speech sampling (Section 5.3) for the ablation study presented in Section 5.5. In all settings Flow Matching with OT paths performs strictly better than both of the other approaches.

## B.6 Transient noise removal in more conditions

We expand the experiments in Section 5.2 by comparing the models on two noise levels (low noise: 10dB and high noise: -10dB), three overlapping ratios (30%, 50%, 70%), and also two types of noise (speech noise and non-speech noise).

Results are presented in Table B6. Voicebox consistently produces the most intelligible audio at all conditions (indicating the percentage of speech to infill). In terms of audio similarity, Voicebox is constantly better in the high noise condition with gains ranging from 0.265 to 0.324 compared to Demucs, and is on par with Demucs in low noise condition.

## B.7 Additional results on inference efficiency versus performance

As explained in Section 5.4, for the cross-sentence setup of Section 5.1, we find that WER remains stable with mean of 2.0 and variance of 0.005. This can be also be observed from Fig. B5a.

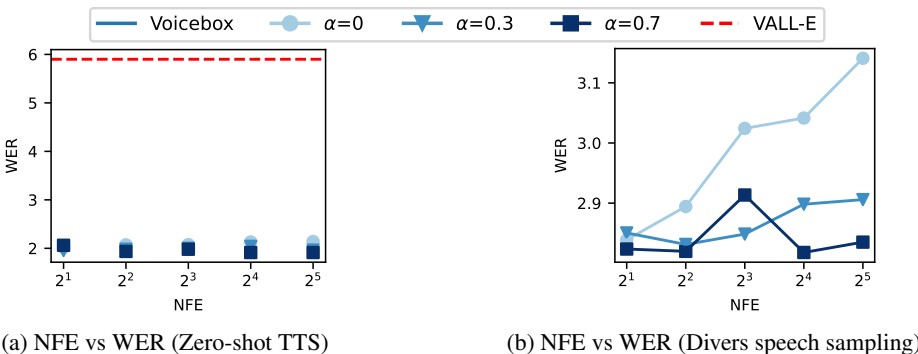

(a) NFE vs WER (Zero-shot TTS)  (b) NFE vs WER (Divers speech sampling)

Figure B5: Trade-off between NFE and WER for different classifier-free guidance strengths (a) presents the WER for cross-sentence zero-shot TTS (Section 5.1) and (b) presents the WER for diverse speech sampling (Section 5.3).

Table B5: Comparison of FM w/OT vs. FM w/Diffusion vs. SM.

| Method | Train Steps | NFE | ZS-TTS (cross-sentence) | | | Diverse sampling | |
| | | | WER | SIM-o | SIM-r | WER | FSD |
|---|---|---|---|---|---|---|---|
| FM w/ OT | 50000 | 4 | 2.7 | 0.303 | 0.362 | 4.8 | 276.499 |
| | | 8 | 2.5 | 0.353 | 0.412 | 4.8 | 235.958 |
| | | 16 | 2.4 | 0.366 | 0.425 | 4.7 | 227.485 |
| | | 32 | 2.5 | 0.364 | 0.424 | 4.7 | 225.931 |
| | 100000 | 4 | 2.5 | 0.347 | 0.404 | 4.3 | 258.358 |
| | | 8 | 2.2 | 0.411 | 0.468 | 4.2 | 216.512 |
| | | 16 | 2.3 | 0.429 | 0.483 | 4.3 | 206.538 |
| | | 32 | 2.2 | 0.431 | 0.487 | 4.2 | 203.792 |
| | 150000 | 4 | 2.4 | 0.356 | 0.410 | 4.0 | 249.712 |
| | | 8 | 2.2 | 0.430 | 0.481 | 4.0 | 208.511 |
| | | 16 | 2.2 | 0.453 | 0.503 | 4.0 | 198.040 |
| | | 32 | 2.1 | 0.458 | 0.508 | 3.9 | 195.304 |
| FM w/ diff | 50000 | 4 | 99.9 | 0.050 | 0.050 | 99.8 | 3478.910 |
| | | 8 | 99.9 | 0.047 | 0.047 | 99.9 | 4704.237 |
| | | 16 | 98.8 | 0.052 | 0.048 | 96.5 | 5336.591 |
| | | 32 | 76.0 | 0.060 | 0.066 | 49.5 | 2485.400 |
| | 100000 | 4 | 98.9 | 0.048 | 0.048 | 96.6 | 4486.401 |
| | | 8 | 14.6 | 0.104 | 0.137 | 12.0 | 669.564 |
| | | 16 | 4.0 | 0.210 | 0.262 | 7.0 | 381.891 |
| | | 32 | 3.1 | 0.285 | 0.344 | 6.3 | 294.777 |
| | 150000 | 4 | 11.5 | 0.132 | 0.171 | 11.4 | 692.560 |
| | | 8 | 3.0 | 0.305 | 0.359 | 5.6 | 334.237 |
| | | 16 | 2.7 | 0.391 | 0.447 | 5.4 | 244.067 |
| | | 32 | 2.6 | 0.423 | 0.478 | 5.2 | 224.963 |
| SM w/ diff | 50000 | 4 | 99.6 | 0.050 | 0.048 | 99.7 | 2816.083 |
| | | 8 | 99.3 | 0.051 | 0.048 | 99.6 | 3079.040 |
| | | 16 | 97.5 | 0.052 | 0.050 | 98.4 | 3710.340 |
| | | 32 | 73.3 | 0.057 | 0.062 | 86.2 | 3011.030 |
| | 100000 | 4 | 99.4 | 0.050 | 0.050 | 99.3 | 3474.579 |
| | | 8 | 97.2 | 0.049 | 0.048 | 97.9 | 3600.423 |
| | | 16 | 53.9 | 0.064 | 0.071 | 69.6 | 2060.892 |
| | | 32 | 17.4 | 0.150 | 0.176 | 34.4 | 1071.579 |
| | 150000 | 4 | 94.5 | 0.055 | 0.054 | 79.4 | 2953.417 |
| | | 8 | 42.3 | 0.070 | 0.076 | 27.5 | 1071.010 |
| | | 16 | 11.5 | 0.191 | 0.218 | 12.8 | 698.411 |
| | | 32 | 5.1 | 0.309 | 0.349 | 8.8 | 519.468 |

In Fig. B5b, we show the WER for the samples generated on Librispeech test-other text. We find that for $\alpha = 0$, WER increases slightly from 2.8 to 3.1 as NFE goes from 2 to 32. For a larger classifier-free guidance strength, WER remains more stable. Subjective listening and FSD reveal that 1) a lower NFE leads to less natural samples with lower diversity, and 2) a higher guidance weight leads to lower diversity. In addition, we observe that ASR can perform well with unnatural samples that contains artifacts, but degrades when the samples are more diverse and expressive (e.g., with whispering voice or with strong reverberation). As a result, we see that the combination of low guidance weight and high NFE leads to a higher WER due to the higher diversity.

## B.8 Choice of audio model output features

The performance of our model is upper bounded by how well the chosen acoustic features can be reconstructed to waveform. The reconstruction performance is determined jointly by the encoding

Table B6: Results of transient noise removal with varying overlapping percentage and noise level. "sp" means added noise is speech, and "non-sp" means non-speech.

| | WER(↓) | | SIM(↑) | | WER(↓) | | SIM(↑) | |
|---|---|---|---|---|---|---|---|---|
| | sp | non-sp | sp | non-sp | sp | non-sp | sp | non-sp |
| | *SNR=-10dB, overlap=30%* | | | | *SNR=10dB, overlap=30%* | | | |
| Noisy speech | 26.7 | 24.9 | 0.202 | 0.238 | 3.7 | 3.1 | 0.605 | 0.603 |
| Demucs | 20.5 | 19.7 | 0.247 | 0.247 | 3.2 | 2.8 | 0.570 | 0.567 |
| A3T | 7.5 | | 0.058 | | *same as left* | | | |
| VB-En ($\alpha = 0.7$) | 2.2 | | 0.566 | | *same as left* | | | |
| | *SNR=-10dB, overlap=50%* | | | | *SNR=10dB, overlap=50%* | | | |
| Noisy speech | 43.6 | 40.8 | 0.256 | 0.292 | 4.5 | 3.8 | 0.649 | 0.649 |
| Demucs | 34.3 | 32.5 | 0.291 | 0.288 | 3.8 | 3.3 | 0.616 | 0.613 |
| A3T | 11.5 | | 0.064 | | *same as left* | | | |
| VB-En ($\alpha = 0.7$) | 2.0 | | 0.612 | | *same as left* | | | |
| | *SNR=-10dB, overlap=70%* | | | | *SNR=10dB, overlap=70%* | | | |
| Noisy speech | 60.0 | 56.0 | 0.260 | 0.303 | 6.3 | 4.6 | 0.595 | 0.592 |
| Demucs | 49.5 | 45.4 | 0.293 | 0.294 | 4.6 | 3.8 | 0.572 | 0.564 |
| A3T | 16.6 | | 0.063 | | *same as left* | | | |
| VB-En ($\alpha = 0.7$) | 2.0 | | 0.559 | | *same as left* | | | |

process, as in how much information is lost when encoding waveform into the features, and the decoding process, as in how well the vocoder can translate the encoded information into waveform.

To motivate the choice of the acoustic feature and the vocoder, we compare four combinations: the first one is Mel spectrogram + HiFi-GAN which is what this paper adopts. The second is Mel spectrogram + Parallel WaveGAN [65] that is used by A3T [3]. The third one is Encodec post-quantization dense feature + Encodec decoder, which is analogous to VALL-E's setup. The last one is also Encodec but with pre-quantization dense feature, which we include to study how much information is lost during quantization.

We also note that Mel spectrogram features are 80 dimensional encoded at 100Hz, which is 8K dimensions per second, while Encodec features are 128 dimensional encoded at 75Hz, which is 9.6K dimensions per second, higher than the Mel spectrogram features.

Table B7 presents the results evaluated on the Librispeech dev-clean and dev-other splits. All three models have the same WER resynthesizing dev-clean split, but ParallelWaveGAN degrades the most on dev-other. Interestingly Encodec even produces audio of lower WER than the ground truth.

In terms of audio similarity, besides the default audio feature extractor WavLM-TDCNN, we also include results of similarity computed with another speaker encoder ECAPA [13]. Parallel WaveGAN is consistently the worst. However, it is unclear whether HiFi-GAN or Encodec performs better. Encodec prevails with the WavLM-TDCNN embedder and HiFi-GAN wins using ECAPA. It may require subjective MOS test to conclude which one reconstructs the audio better, and we leave exploration of modeling Encodec dense features for future study.

Table B7: Comparison of different audio features and vocoders on audio reconstruction. Librispeech dev-clean (d-c) and dev-other (d-o) are used for evaluation. WER and audio similarity computed with WavLM-TDCNN and ECAPA are reported.

| Audio feature / Vocoder | WER | | SIM-o (WavLM) | | SIM-o (ECAPA) | |
|---|---|---|---|---|---|---|
| | d-c | d-o | d-c | d-o | d-c | d-o |
| Ground truth | 2.1 | 4.7 | 1.000 | 1.000 | 1.000 | 1.000 |
| Mel spectrogram / HiFi-GAN | 2.1 | 4.7 | 0.915 | 0.909 | 0.766 | 0.762 |
| Mel spectrogram / Parallel WaveGAN | 2.1 | 5.2 | 0.868 | 0.847 | 0.721 | 0.711 |
| Encodec post-quantized feature / Encodec decoder | 2.1 | 4.5 | 0.943 | 0.944 | 0.724 | 0.722 |
| Encodec pre-quantized feature / Encodec decoder | 2.1 | 4.4 | 0.943 | 0.944 | 0.724 | 0.722 |

# C   Additional Details and Studies on Metrics

## C.1   Measuring speech diversity and quality with FSD

**Diversity**   We first validate if FSD reflects the diversity for a set of speech samples and study its sensitivity to sample size. To achieve that, we design controlled experiments to compute FSD on sets of samples with varying diversity and sample sizes. Specifically, we create two partitions from 1K hours of English speech, where each partition has the same set of speakers and the same number of utterances for each speaker. The first partition is considered the reference set.

To test the sensitivity to sample size, we use the second partition to create subsets by sampling $r\%$ of utterances from each speaker in that partition. This sampling method is denoted as "utt". We computed that on average, each speaker contributed approximately $2.33$ sessions, with each session containing around $52.45$ utterances. Therefore, the subsets created using the sampling method are expected to have similar audio style distributions to the reference set and the FSD is expected to stay low regardless of the subset size. We consider $r \in \{1, 5, 10, 25, 50, 100\}$.

To test the correlation with diversity, we again use the second partition to create subsets by sampling $r\%$ of speakers and including all the utterances in the partition from those speakers. This sampling method is denoted as "spk" where a smaller $r$ leads to a subset with fewer speakers and hence lower diversity. Therefore the FSD is expected to increase as $r$ decreases. The same set of values for $r$ is considered. For the same $r$, the "utt" subset should always have a lower FSD than the "spk" subset.

We compare three different features for computing the FSD score. The first is the supervised WavLM-TDCNN feature used for computing audio similarity (SIM-r and SIM-o). The second is the self-supervised wav2vec 2.0 BASE [2] feature reduced to 128 dimensions using principle component analysis (PCA). The last one is the supervised audio event classification model feature that is used to compute FAD [30] for non-speech audio generation.

Figure C6 first compares using different layers of wav2vec 2.0 features. All of them yield similar desirable results where "utt" stays low and "spk" increases drastically when the sample size reduces and speaker diversity decreases. We then decide to use the middle layer (layer 6) as the default feature for FSD computation.

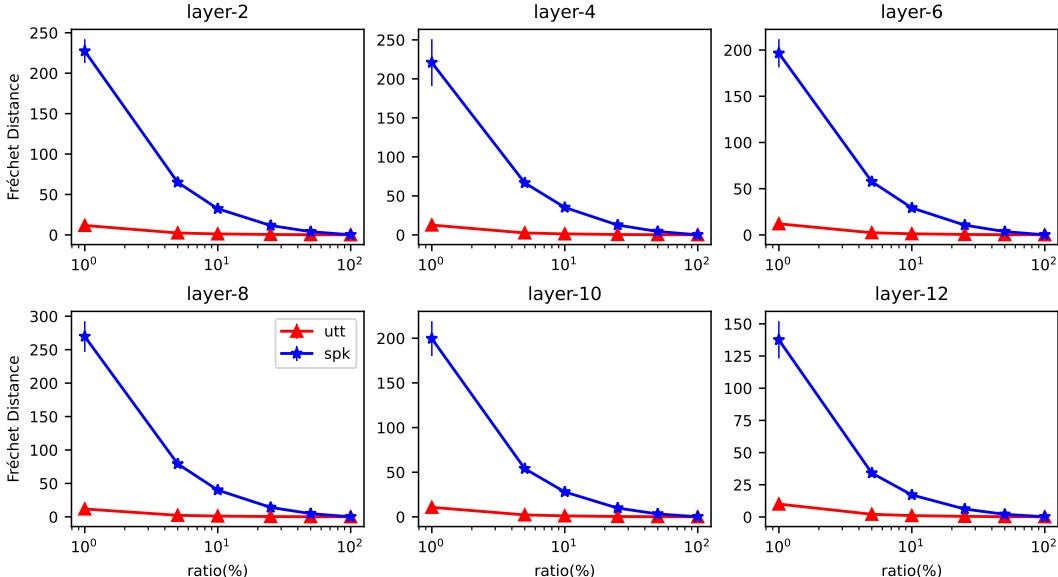

Figure C6: FSD based on different layers of wav2vec 2.0 BASE. utt: utterance-based sampling, spk: speaker-based sampling. Vertical bars denote standard deviation.

Figure C7 further compares wav2vec 2.0-layer 6 with the two other features. WavLM-TDCNN and wav2vec 2.0-layer 6 present similar trends and both have low variance. Both of them are suitable for measuring diversity, and we decide to use wav2vec 2.0 features as it is self-supervised and would be able to capture more holistic information of speech such as prosody and emotion.

In contrast, FAD score [30] is not appropriate for measuring speech diversity. The score does not increase much between $r = 25\%$ and $r = 1\%$ for "spk" sampling method, showing that the score does not reflect the decreasing speaker diversity. On the other hand, "utt" sampling method observes huge FAD score increase when reducing the sample size from $r = 25\%$ to $r = 1\%$ where the diversity does not change much as the number of speakers remains the same. Moreover, at $r = 1\%$ both sampling methods result in similar FAD score while the two subsets exhibit very different levels of diversity. We hypothesize that this is because FAD score is computed based on features extracted from an audio even classifier trained on AudioSet, which learns to distinguish between events like lawn mower, car engine, and human speech, but does not learn to capture the variation within speech, such as different voices.

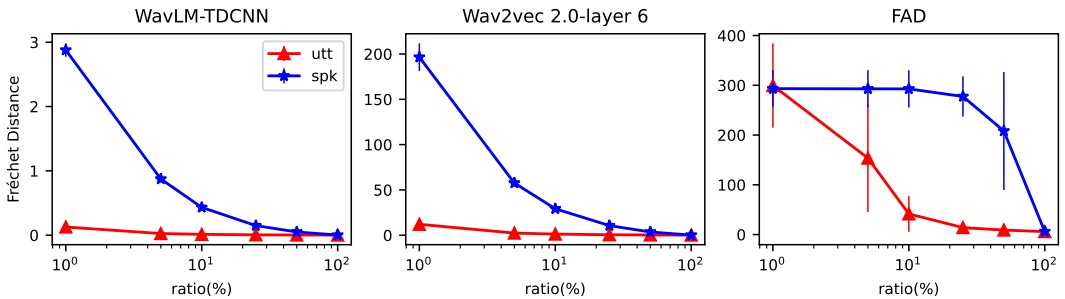

Figure C7: FSD with different sample size using supervised WavLM-TDCNN, self-supervised wav2vec 2.0, and supervised audio event classifier features. utt: utterance-based sampling, spk: speaker-based sampling. Vertical bars denote standard deviation.

**Quality**    In addition to measuring diversity, Fréchet distance is a commonly used metric for assessing quality in image generation [19]. To show its applicability for speech generation, we evaluate the FSD score of speech utterances with varying levels of quality. The reference set samples are 1K hours of English training data, and the hypothesis set is the Librispeech test-clean split with noise added. We added Gaussian noise at different SNRs, ranging from 0 to 50 dB. Lower SNR values correspond to lower quality. We use the default speech feature extractor (i.e., wav2vec 2.0, layer-6) throughout the experiments.

Our results, summarized in Figure C8, show that a subset with a lower SNR has a higher FSD score. Therefore, a lower FSD score indicates higher acoustic quality for the set of test samples when diversity is fixed.

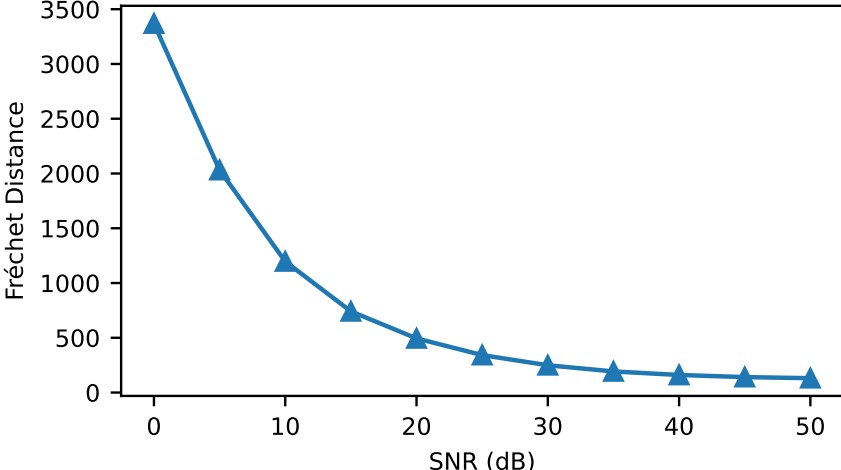

Figure C8: FSD under different noisy levels. Feature: Wav2vec 2.0 layer-6 feature. Noise is added upon model output from Voicebox under unconditional setting.

## C.2 Standalone metrics for duration models

As mentioned in the main text, we can utilize end-to-end metrics of WER, SIM, and FSD to evaluate duration models, but also consider metrics specifically for duration.

First, we consider two metrics aimed at the quality of duration predictions, here denoted $\hat{l}(l_{ctx}, y)$. For a regression model, we use $\hat{l}(l_{ctx}, y) = g(l_{ctx}, y; \theta)$. For a flow matching model, we set $\hat{l}$ as the mean over 20 samples, ensuring a fairer comparison.

**Duration correctness (MS-MAE)**   Our first metric, multi-sample mean-absolute error (MS-MAE), is the masked absolute error per-utterance divided by the average number of masked phonemes per-utterance

$$\frac{\mathbb{E}_{m,l,y}||m \odot \left( l - \hat{l}(l_{ctx}, y) \right) ||_1}{\mathbb{E}_{m,l,y}||m||_1} \tag{4}$$

**Speaking rate correlation (MS-Corr)**   Our next metric, multi-sample correlation (MS-Corr), computes the average masked predicted duration and unmasked duration context per utterance, and computes their correlation across utterances. Comparing MS-Corr with the same correlation computed from the ground truth, we observe to what extent predicted durations capture appropriate correlations with the context.

**Duration diversity and quality (FDD)**   Additionally, we evaluate the quality and diversity of duration samples at the distribution level, similar to our audio evaluation of diversity and quality via FSD. We produce one sample per utterance from a duration model and collect all sampled phoneme durations, possibly many per-utterance, into an empirical distribution. We compare means and variances of this sampled distribution versus the means and variances of the training distribution, labeled $\mu$, $s$, and $\mu'$, $s'$ respectively. We define the Fréchet duration distance (FDD) as the Fréchet distance between the distributions

$$(\mu - \mu')^2 + s + s' - 2\sqrt{ss'}, \tag{5}$$

treated as though they were Gaussians. FDD depends on the sampled durations accurately reflecting the training distribution of real durations. As for FSD, this metric is specific to unconditional text-to-speech generation.

## C.3 Duration model evaluation with standalone metrics

We evaluate three duration model variants. The first and second utilizes flow matching and regression, trained using masked conditional flow matching and regression respectively as described in Section 3.3. The third is a regression model that ignores duration context $l_{ctx}$ and only uses phonetic transcript $y$, referred to as unconditional regression below. This is the duration model used in FastSpeech2 [51], A3T [3] and many other non-autoregressive speech synthesis models.

We evaluate our three duration model variants on the Librispeech test-other split on two tasks. The first is unconditional TTS where we generate all durations from given phonemes (i.e. $l_{ctx}$ is entirely masked). The second task is infilling the second half of each utterance's durations, where $l_{ctx}$ are durations from the unmasked half of the utterance. This second infilling task distinguishes between the two regression model variants, since the unconditional regression ignores $l_{ctx}$, and hence predicts identical durations for the tasks. Duration metrics are computed for TTS and infilling in Table C8 and C9. The prefix Phn or Sil indicates the associated metric was either computed across all non-silence or all silence phonemes. Start and end silences were not trimmed for these duration metric evaluations.

Starting with prediction quality metrics (MS-MAE and MS-Corr), the duration-conditional regression performs slightly better on MS-MAE overall than the other models. Larger differences are seen on Phn-MS-Corr where the unconditional regression has a correlation substantively below the other models (Phn-MS-Corr of ground truth is $0.47$), indicating conditioning on duration context $l_{ctx}$ is beneficial. Flow-matching shows the largest distinction versus regression on the distributional comparison captured by FDD. The regression models have generally larger FDD because they underestimate the standard deviation in phoneme and silence durations, and hence produce samples with less duration diversity and more regular duration lengths.

Table C8: English TTS duration metrics on LS test-other.

| Duration Model | Phn-MS-MAE | Phn-FDD | Sil-MS-MAE | Sil-FDD |
|---|---|---|---|---|
| Unconditional Regression | 2.53 | 0.72 | 5.32 | 2.39 |
| Duration-conditional Regression | 2.52 | 0.76 | 5.10 | 8.40 |
| Duration-conditional Flow Matching | 2.63 | 0.61 | 5.18 | 2.48 |

Table C9: English second-half infilling duration metrics on LS test-other.

| Duration Model | Phn-MS-MAE | Phn-MS-Corr | Sil-MS-MAE |
|---|---|---|---|
| Unconditional Regression | 2.57 | 0.26 | 5.44 |
| Duration-conditional Regression | 2.45 | 0.35 | 5.20 |
| Duration-conditional Flow Matching | 2.52 | 0.41 | 5.32 |

## C.4 Duration model evaluation with end-to-end metrics

We now present end-to-end metrics for our three duration variants for zero-shot TTS cross-sentence and continuation, as well as diverse speech generation, corresponding to Sections 5.1 and 5.3. Zero-shot TTS cross-sentence and continuation results are shown in Table C11 and diverse speech generation results in Table C10. These results are not comparable with the main text as they utilize the flow-matching model described in Appendix B.3, denoted as VB-En-1K.

Overall, FSD and SIM are similar across duration variants. On the other hand, WER is sensitive to the choice of duration model, where the duration-conditional regression achieves a substantially lower WER. Subjective listening from the duration-conditional regression and flow-matching confirms that the regression model is producing more regular patterns of speech, that may be easier for ASR to recognize, while sacrificing some duration diversity.

Table C10: Diverse speech generation from LS test-other text.

| Duration Model with VB-En-1K | WER | FSD (LS-train) |
|---|---|---|
| Unconditional Regression | 3.8 | 148.7 |
| Duration-conditional Regression | 3.7 | 148.1 |
| Duration-conditional Flow Matching | 5.4 | 155.1 |

## C.5 MOS instructions

Table C12 shows the instruction presented to the raters for quality mean opinion score study. Table C13 shows the instruction presented to the raters for similarity mean opinion score study.

# D Limitation and Broader Impact

**Limitation** Voicebox models presented in this paper are trained on read speech from audiobooks in up to six written languages. Hence, the current models may not transfer well to conversational speech [15], which is more casual and contains more non-verbal sounds such as laughing and back-channeling (e.g., um-hmm). We plan to tackle the problem by scaling the training data to incorporate more diverse speech.

On the other hand, Voicebox depends on a phonemizer and a forced aligner to produce frame-level phonetic transcript. In addition, many existing phonemizers [40] are word-based, which does not take neighboring words of the target into account when predicting the pronunciation. Such phonemizers cannot accurately predict phonetic transcript given text because pronunciation is context-dependent in many languages (e.g., liaisons in French). In the future, we will explore more end-to-end methods where a model would be able to take raw text with punctuation as input [7], and eliminate the need of phonemizers and forced aligners to improve the performance and increase the language coverage.

Table C11: English zero-shot TTS results on filtered LS test-clean.

| Duration Model with VB-En-1K | WER | SIM-o | SIM-r |
|---|---|---|---|
| *cross-sentence* | | | |
| Unconditional Regression | 3.0 | 0.538 | 0.584 |
| Duration-conditional Regression | 2.7 | 0.545 | 0.591 |
| Duration-conditional Flow Matching | 3.4 | 0.528 | 0.578 |
| *continuation* | | | |
| Unconditional Regression | 2.5 | 0.485 | 0.524 |
| Duration-conditional Regression | 2.2 | 0.491 | 0.533 |
| Duration-conditional Flow Matching | 2.7 | 0.481 | 0.525 |

Table C12: Quality mean opinion score (QMOS) instruction.

**Introduction**
Your task is to evaluate the subjective quality and intelligibility of the speech from short (2-8 second) audio files. Each HIT can be completed in roughly around 120 seconds.

**Task Instructions**
In this task you will hear samples of speech recordings. The purpose of this test is to evaluate the quality and intelligibility of each file in terms of its overall sound quality and the amount of mumbling and unclear phrases in the recording.

Please keep in mind that speech samples can be distorted and noisy, however these are only specific examples.

Please use a headset for listening and adjust your volume level to your comfort during this training, and do not change later during the experiment.

You should give a score according to the following scale, known as the MOS (mean opinion score) scales:

**Score (Quality and Intelligibility of the speech)**
5 (Excellent)
4 (Good)
3 (Fair)
2 (Poor)
1 (Bad)

Last but not least, while Voicebox yields impressive results on transferring audio style (voice, speaking style, emotion, and acoustic condition), the model does not allow independent control of each attribute. In other words, one cannot ask the model to generate speech that resembles voice of one sample while resembling the emotion of another sample. We leave disentangled control of attributes through prompting or text description for future work.

**Broader impact**   A high-quality and versatile generalist speech generation model like Voicebox can enable many applications that improve the quality of our life. For example, zero-shot TTS could bring the voice back to people who suffer from diseases or underwent surgeries such as laryngectomy the causes inability to speak. Zero-shot TTS can also be combined with visual speech recognition systems [23] to avoid the need of typing. When paired with speech translation models, cross-lingual zero-shot TTS enables everyone to speak any language in their own voice. Content editing and speech denoising can be productivity tools for users to create content more effortlessly. Diverse speech sampling, as shown in the paper, can significantly reduces the cost of creating data for training speech-input models.

While Voicebox can bring many positive social impacts, it also carries the potential of misuse and unintended harm. To mitigate the risk, we have presented a highly effective classifier showing that the

Table C13: Similarity mean opinion score (SMOS) instruction.

**Task Name**
Rate the similarity of the synthesized speech samples to a given prompt.

**Task Instructions**
Your task is to evaluate the similarity of the synthesized speech samples to the given speech prompt. You should focus on the similarity of the speaker, speaking style, acoustic conditions, background noise, etc. You should rank the recordings on the scale between 1-5, where 5 is the best quality and 1 is the worst.

In other words, please rank the recordings according to their acoustic similarity to the given prompt, meaning as if they were recorded in the same place by the same speaker speaking in similar styles. This task typically requires approximately 120 seconds to complete.

Please use a headset for listening and adjust your volume level to your comfort during this training, and do not change later during the experiment.

---

model can accurately distinguish between real and synthetic speech. For future work, we also plan to investigate proactive methods for training the generative model such that the synthetic speech can be more easily detected, such as embedding artificial fingerprints [66] that can be trivially detected without hurting the speech quality.

To prevent Voicebox from learning biases, we also need to carefully select its training data. First, if Voicebox is only trained on a smaller number of speakers from a specific group with similar accents, it will not be able to generate diverse speech representing the accents around the globe, and downstream models trained on Voicebox generated speech would perform worse on groups with underrepresented accents. For zero-shot style transfer, the performance would also degrade for underrepresented accents. To mitigate this, we have leveraged in-the-wild speech that includes a wide variety of accents, and will continue investing in collecting diverse speech to avoid such biases.

Second, if Voicebox is trained on data where samples from one ethnic group always have lower audio quality (e.g., more noise) while the other ethnic group always has higher audio quality samples, the model would also learn undesired association. To mitigate this, we want the distribution of audio quality (and other audio attributes) and ethnic group to be less correlated, which is usually the case when we have larger scale data collected from in-the-wild sources. We can further tackle this by leveraging data augmentation to decorrelate the distribution, such as adding noise and enhancing speech to widen the audio quality distribution.

