# OpenReview forum: "Voicebox: Text-Guided Multilingual Universal Speech Generation at Scale"
_NeurIPS.cc/2023/Conference — NeurIPS 2023 poster_

### Official Review · Reviewer_MQeA · 2023-06-29

**Soundness:** 2 fair
**Presentation:** 3 good
**Contribution:** 1 poor
**Rating:** 5
**Confidence:** 5

**Summary:**

This work presents a text-conditional speech synthesis model with an optimal transport path and conditional flow matching. The model is trained with a large-scale cross-lingual dataset for zero-shot style transfer and content editing.

**Strengths:**

The work adopts a conditional normalizing flow with an optimal transport path for speech synthesis. The proposed model can synthesize and edit the speech with different styles.

**Weaknesses:**

Although this work successfully executes the zero-shot TTS and speech editing, I have doubts on the originality of this work and I think that the authors should have conducted more comparisons with the recently proposed TTS models (not YourTTS, not Vall-E), and speech editing or speech correcting paper.

I have some comments on this work.

1. The concept to train the model by infilling speech given audio and text has been already presented in many works. First, SpecAugment [1] presented a time masking for context learning and Conformer [2] successfully adopted it for the ASR task. Self-supervised speech representation also models such as wave2vec 2.0 successfully learned the context with masking. Moreover, there are many similar models which can edit the speech conditional text information in speech editing [3] [4], correcting domains [5] [6], and also in the TTS domain [7] [8]. The authors only adopt the conditional normalizing flow with an optimal transport path for text-to-speech.

[1] Park, Daniel S., et al. "Specaugment: A simple data augmentation method for automatic speech recognition." Interspeech, 2019.

[2] Gulati, Anmol, et al. "Conformer: Convolution-augmented transformer for speech recognition." Interspeech, 2020.

[3] Tae, Jaesung, Hyeongju Kim, and Taesu Kim. "EdiTTS: Score-based editing for controllable text-to-speech." Interspeech, 2022.

[4] Wang, Tao, et al. "Campnet: Context-aware mask prediction for end-to-end text-based speech editing." IEEE/ACM Transactions on Audio, Speech, and Language Processing 30 (2022): 2241-2254.

[5] Tan, Daxin, et al. "CorrectSpeech: A Fully Automated System for Speech Correction and Accent Reduction." ISCSLP, 2022.

[6] Fong, Jason, et al. "Speech Audio Corrector: using speech from non-target speakers for one-off correction of mispronunciations in grapheme-input text-to-speech." Proc. Interspeech 2022 (2022): 1213-1217.

[7] Ao, Junyi, et al. "Speecht5: Unified-modal encoder-decoder pre-training for spoken language processing." ACL, 2022.

[8] Wang, Tao, et al. "Non-Autoregressive End-to-End TTS with Coarse-to-Fine Decoding." INTERSPEECH. 2020.

2. Specifically, the diffusion-based model (EdiTTS [3] and Guided-TTS [9]) can edit the speech even without training. The author should have compared the conditional normalizing flow with a diffusion-based model. However, the authors only referred that OT path leads to faster training, faster generation, and better performance compared to diffusion paths. I wonder if the conditional normalizing flow with OT path is better than diffusion-based models.

[9] Kim, Heeseung, Sungwon Kim, and Sungroh Yoon. "Guided-tts: A diffusion model for text-to-speech via classifier guidance." ICML, 2022.

3. The audio quality is not good on the demo pages.

4. For a fair comparison, all models should be trained with the same dataset.

5. YourTTS is not a good text-to-speech model for zero-shot text-to-speech models. The audio quality of YourTTS is bad. I recommend training the VITS with the reference encoder you used and the same configuration. In addition, transferring the voice style from a reference audio, not speaker ID, is utilized in many works. For a fair comparison, the same transferring method is used for each model.

**Questions:**

1. The authors utilized a 100 Hz frame rate to extract the high-resolution Mel-spectrogram. For a fair comparison, each model is also trained with the same time resolution for Mel-spectrogram. For example, VITS and YourTTS should be trained with the linear spectrogram extracted at a 100 Hz frame rate for high-quality. However, the details are not described for baseline models.

2. Are there any scenarios for automatic noise removal or editing? In this work, the location of audio for noise removal or editing should be segmented by the user. [5] presented the fully automated speech correction scenario with detection, correction, and generation. It would be nice incorporating ASR with this work for an automatic speech editing system.

3. I think replacing the HiFi-GAN with BigVGAN could improve the audio quality.

4. For the AR model, the dropout in the pre-net of the decoder may improve the diversity of speech. Also, VITS and YourTTS can increase the sample diversity by controlling the temperature T. The details you used should be described.

**Limitations:**

They stated the limitations and potential negative societal impact on Section Conclusion.

---

> ### Author Rebuttal · Authors · 2023-08-08
>
> We thank the reviewer for their thoughtful feedback. We address common questions in Author Rebuttal above and other questions here. We will incorporate all feedback in the final version.
>
> **1. The concept to train the model by infilling speech given audio and text has been already presented in many works. SpecAugment [1] presented a time masking for context learning and Conformer [2] successfully adopted it for the ASR task. Self-supervised speech representation also models such as wave2vec 2.0 successfully learned the context with masking.**
>
> We would like to emphasize that this work focuses on building a **generalist speech generation model**. In contrast, **SpecAugment, Conformer, and wav2vec 2.0 are ASR/representation learning models that do not infill speech and cannot generate speech. They should not be considered related work.** [1] and [2] use masking for regularization. wav2vec 2.0 is similar to CPC which masks to create different views for contrastive learning.
>
> **2. there are many similar models which can edit the speech conditional text information in speech editing [3] [4], correcting domains [5] [6], and also in the TTS domain [7] [8]. The diffusion-based model (EdiTTS [3] and Guided-TTS [9]) can edit the speech even without training**
>
> **CampNet [4] is very similar to A3T, which we have already compared with.** Both assume a deterministic input/output mapping, preventing speech infilling to generalize to longer spans. The key differences are [4] does not require alignment during training, and uses a two-stage NAR decoder to refine spectrogram.
>
> **[3-9] are all very different from Voicebox and cannot be compared in the same setup.** [3] and [6] require an audio sample containing the new text to be swapped in for editing. To edit “I’m happy” into “I’m heavy”, [3] requires an audio sample of the same speaker saying “heavy”, while [6] requires a sample of “heavy” that can be from a different speaker. [5] presents an automatic way to align source text with target text and reuse [4] for editing, which is complementary but orthogonal to Voicebox.
>
> SpeechT5 [7] is an unsupervised pre-training framework, which requires separate fine-tuning to perform each task, and cannot edit speech or perform in-context style transfer. [8] is a regression-based NAR TTS model evaluated on a 20-hour single speaker dataset and does not edit speech. It will suffer similar issues as A3T when trained on diverse data. [9] did not show that it is capable of editing..
>
> **3. each model trained with the same time resolution for Mel-spectrogram. For example, VITS and YourTTS should be trained with the linear spectrogram extracted at a 100 Hz frame rate for high-quality.**
>
> **What the reviewer suggests is similar to Glow-TTS+ HiFi-GAN. We believe VITS serves as a stronger baseline for flow-based TTS,** given VITS has shown that it is better (Table 1 and 3 in VITS). The frame rate of Glow-TTS is 22K / 256 = 86Hz which is close to the 100Hz we used. Moreover, higher time resolution does not necessarily mean better quality, it depends on if a model has enough model capacity.
>
> **4. For the AR model, the dropout in the pre-net of the decoder may improve the diversity of speech. Also, VITS and YourTTS can increase the sample diversity by controlling the temperature T.**
>
> **Pre-net dropout, which serves for regularization, can lead to stochastic behavior but the output diversity is limited.** Moreover, AR models like Tactoron still make an overly strong conditional independence assumption where each feature dimension is conditionally independent for a given time step.
>
> For VITS trained on diverse speech without a speaker encoder, one can draw samples from the low-dimensional Gaussian prior to generate diverse samples. We have presented results of this in the common comment 3 above.
>
> We follow the recommended temperature setup for YourTTS for sampling. It can be seen in Table 5 and 6 that the gap to Voicebox is huge (FSD: 277.9 vs 159.8/test-o WER: 54.6% vs 8.3%). YourTTS also conditions generation on speaker embedding inferred from a reference audio, and hence it cannot perform diverse sampling without conditioning on any audio like Voicebox.
>
> **5. The audio quality is not good on the demo pages.**
>
> All the samples presented in the supplementary material use audio prompts from recruited volunteers, who recorded samples from their own devices and provided consents for sharing. Hence it can be heard that some prompts also contain noise and are not high quality. The quality of the produced audio samples should be compared against the quality of the audio prompt (“Voicebox Input”, “Original Speech”, and “Prompt”). This is because the Voicebox transfers audio style from the prompt, including not only voice, but also audio quality (noise, reverberation, etc). We would be more than happy to discuss if the reviewer could point out any specific samples with noticeable worse quality compared to their input audio prompt.
>
> **6. Are there any scenarios for automatic noise removal or editing?**
>
> This paper did not explore automatic methods for determining noisy segments. One can consider using reference-free quality estimation methods like torchaudio-squim [1] or WADA-SNR [2] to determine which segments are noisy and should be re-generated. For editing, CorrectSpeech is applicable to not only CampNet but also A3T and Voicebox.
>
> [1] Kumar et al. "Torchaudio-Squim: Reference-Less Speech Quality and Intelligibility Measures in Torchaudio." ICASSP’23
>
> [2] Kim and Richard. "Robust signal-to-noise ratio estimation based on waveform amplitude distribution analysis." Interspeech’08
>
> **7. replacing the HiFi-GAN with BigVGAN could improve the audio quality.**
>
> We thank the reviewer for their suggestion and agree that BigVGAN would likely lead to better audio quality. We will explore in the future work, and we also want to note that our main contribution is orthogonal to the choice of the vocoder.

---

> > ### Comment · Reviewer_MQeA · 2023-08-14
> > **Thanks for your response**
> >
> > Thank you for your helpful response.
> >
> > I acknowledge that this work first proposes the generalized model for multiple speech tasks. Moreover, I think your model will have a good impact on speech research. That is why the authors should conduct more experiments fairly for future researches who might follow your research. I don't want anyone in the future to claim "VoiceBox experimented like this and we're just following VoiceBox".
> >
> > I still have a concern about the comparisons for each task. I hope that you do not cut corners by just arguing that our framework is novel.
> >
> > >**Zero-shot TTS experiment**
> >
> > I still disagree with your experiments because YourTTS is not a good zero-shot TTS model. The audio quality is not good and the models are trained with different datasets. It is not fair to compare it with your model. The authors also utilize an additional Duration model.
> >
> > >**Diverse speech generation experiment**
> >
> >  you just compared the model with VITS-LJ and VITS-VCTK. I could not agree that your model is better than others with the results of this experiment. You should train the model with the same dataset for a fair comparison.
> >
> > >**To generalize speech infilling, any powerful non-autoregressive generative models, including diffusion models, should work. We chose flow-matching (FM) with optimal transport (OT) path because [1] showed that FM w/ OT > FM w/ diffusion > score-matching (SM) w/ diffusion (the typical diffusion model) on training speed and inference compute-quality trade-off. See the comparison in Table 1 and Fig 4-7 in [1].**
> >
> > If you're saying this, adopting the flow-matching (FM) with optimal transport (OT) path is not your contribution. the authors should have compared all scenarios (FM w/ OT, FM w/ diffusion, score-matching (SM) w/ diffusion) to verify that the FM w/ OT is also a better method for speech generative tasks.
> >
> > > **Duration Modeling**
> >
> > I also have an additional doubt on the comparison of different models. In Appendix, the WER results of flow-matching and regression model are almost same in Table B3. This results show that the trained model with your dataset has a just lower WER so you should train the VITS or other models with the same dataset you used. I think a duration modeling with a large-scale dataset improves the pronunciation. For a fair comparison, VITS with a duration modeling should be compared. Specifically, VITS just utilizes a MAS for a efficient training without external duration modeling. In addition, there are many works which utilizes VITS with external duration modeling to improve the performance.
> >
> > [1] Cite as: Ju, Y., Kim, I., Yang, H., Kim, J.-H., Kim, B., Maiti, S., Watanabe, S. (2022) TriniTTS: Pitch-controllable End-to-end TTS without External Aligner. Proc. Interspeech 2022, 16-20, doi: 10.21437/Interspeech.2022-925
> >
> > [2]   Cite as: Lim, D., Jung, S., Kim, E. (2022) JETS: Jointly Training FastSpeech2 and HiFi-GAN for End to End Text to Speech. Proc. Interspeech 2022, 21-25, doi: 10.21437/Interspeech.2022-10294
> >
> > [3] Zhang, Yongmao, et al. "Visinger: Variational inference with adversarial learning for end-to-end singing voice synthesis." ICASSP 2022-2022 IEEE International Conference on Acoustics, Speech and Signal Processing (ICASSP). IEEE, 2022.
> >
> > [4] Y. Shirahata, R. Yamamoto, E. Song, R. Terashima, J. -M. Kim and K. Tachibana, "Period VITS: Variational Inference with Explicit Pitch Modeling for End-To-End Emotional Speech Synthesis," ICASSP 2023 - 2023 IEEE International Conference on Acoustics, Speech and Signal Processing (ICASSP), Rhodes Island, Greece, 2023, pp. 1-5, doi: 10.1109/ICASSP49357.2023.10096480.
> >
> > Basically, I like the concept of this paper. However, current manuscript does not conducted a fair comparison. I encourage the authors to add additional ablation studies to the paper. But, they did not conduct any experiments I suggested so I could not take any action in this stage.

---

> ### Author Response · Authors · 2023-08-18
> **Thank you for your follow-up comments (1/2)**
>
> We thank the reviewer for carefully reading our response and sharing additional comments. We are glad that the reviewer acknowledged the novelty of Voicebox and its positive impact for future speech research.
>
> We wholeheartedly agree with the reviewer that proper comparisons are required and conclusions should be drawn carefully. **Respectfully, we disagree with the reviewer’s comments on “current manuscript does not conduct a fair comparison” and “[the authors] did not conduct any experiments [the reviewer] suggested.”** In our initial author response, we have pointed out where some of the suggested experiments can be found in the original manuscript and have added additional ablation studies. We have also kindly asked for clarification for one experiment the reviewer suggested (train VITS with same reference encoder), because the initial suggestion was not feasible. Unfortunately, we have not received clarification and are not able to conduct that experiment.
>
> We address each comment with our itemized response below.
>
> ---
>
> > **Comment 1:** [Diverse speech generation experiment] you just compared the model with VITS-LJ and VITS-VCTK. I could not agree that your model is better than others with the results of this experiment. You should train the model with the same dataset for a fair comparison.
>
> - **This is not true.** We also compared with (a) YourTTS trained on LibriTTS + 2 others, (b) A3T trained on VCTK, (c) Voicebox trained on 1K hour audiobooks, (d) Voicebox with A3T objective trained on 1K hour audiobooks, (e) VITS trained on 1K hour audiobook
>
> - (a) and (b) are presented in the paper Table 5 and 6, (c) and (d) are presented in the appendix Table B3, (e) is presented in the rebuttal “Global Author Response”, Comment 3.
>
> -  **(c), (d), (e) are all trained with the same dataset** and we can draw the conclusion that Voicebox is the best in that controlled setup.
>
> ---
>
> > **Comment 2:** [Zero-shot TTS experiment] I still disagree with your experiments because YourTTS is not a good zero-shot TTS model. The audio quality is not good and the models are trained with different datasets. It is not fair to compare it with your model. The authors also utilize an additional Duration model.
>
> - **YourTTS and VITS also have a duration model** that is jointly trained (Sec 2.2.2. in VITS). Hence, Voicebox does not utilize an additional duration model.
>
> - As pointed out in Global Author response (comment 3), we believe **comparing Voicebox and VALL-E is fair** as it meets reviewer's criteria: Both are trained on the same dataset and adopt the same style transfer method. VALL-E also represents the most recent and the strongest baseline on ZS-TTS.
>
> - **The reviewer initially suggested that we should compare with other baselines instead of VALL-E, despite that VALL-E meets all the criteria listed.** We have kindly asked the reviewer to provide details on this comment for us to better address the concern. Unfortunately we did not receive clarification.
>
> - **It is unclear from the reviewer what would constitute a fair comparison between Voicebox and YourTTS/VITS.**
>   - **YourTTS uses a pre-trained speaker embedder**. It still would not be a fair comparison even if we train YourTTS on the same dataset.
>   - **We have trained a vanilla VITS on the same dataset** but it is not capable of zero-shot TTS. We compared it on the diverse speech generation task.
>   - As explained in “Global Author Response”, Comment 3, **Voicebox does not have an explicit encoder so the suggestion of “training VITS with [our] reference encoder” is not feasible. We have kindly requested clarification but did not receive responses.** Moreover, **such a model, even if feasible, already deviates from the original VITS/YourTTS.** It changes not only the training data, but also the task (from TTS to text-guided infilling) and the model architecture. At that point, it should not be considered as a comparison with an existing baseline.
>
> - **We present a new experiment here to address the concern on the comparison with YourTTS in terms of data.** We trained Voicebox on LibriTTS with the reduced setup described in appendix Section B3, which is strictly a subset of what YourTTS is trained on (LibriTTS, VCTK, PT, FR SCL). Results suggest that Voicebox are still significantly better
>
> | Model   | ZS-TTS WER (lower is better) | ZS-TTS SIM-o (higher is better) |
> | -------- | ------- | ------- |
> | Voicebox (trained on LibriTTS, reduced setup) | 2.1% | 0.579 |
> | YourTTS | 7.7% | 0.337 |

---

> ### Author Response · Authors · 2023-08-18
> **Thank you for your follow-up comments (2/2)**
>
> > **Comment3:** In Appendix, the WER results of flow-matching and regression model are almost same in Table B3. This results show that the trained model with your dataset has a just lower WER so you should train the VITS or other models with the same dataset you used.
>
> - **We highlight that the speaker similarity has a big gap between the flow-matching model and the regression model** (0.597 vs 0.520 SIM-r, the higher the better), and similarly for the sample diversity (242.5 vs 278.8 FSD, the lower the better).
>
> - **We have trained a vanilla VITS on the same dataset and presented the diverse speech generation results in our initial response** (“Global Author Response”, Comment 3). The WER is 16.6% and FSD is 311.75.
>
> ---
>
>
> > **Comment4:** If you're saying this, adopting the flow-matching (FM) with optimal transport (OT) path is not your contribution. the authors should have compared all scenarios (FM w/ OT, FM w/ diffusion, score-matching (SM) w/ diffusion) to verify that the FM w/ OT is also a better method for speech generative tasks.
>
> - Exactly because this is not our main contribution, we focus on comparing our method (a gradient-based NAR generative model) with token-based autoregressive models (VALL-E) and with regression-based non-autoregressive models (A3T, CampNet), and highlighting the contrast in task generalization, performance, and inference efficiency among these model families.
>
> - **We present new experiments here comparing the three gradient-based methods, and confirm FM w/ OT indeed has the best training and inference efficiency.** We adopt the same ablation setup as Appendix Section B.3 (with loss computed on all frames and a smaller learning rate, 1e-4, to ensure convergence for all three methods). We vary the number of training and inference steps
>
> Exp 1: Training for 50K / 100K / 150K updates; Inference with 64 NFEs. **We see FM w/ OT achieves the best performance with 100K training steps, and even outperforms SM w/ diff using only 50K updates.**
>
> | Model   | ZS-TTS WER (upd=50K/100K/150K) | ZS-TTS SIM-o (upd=50K/100K/150K)  |
> | -------- | ------- | ------- |
> | FM w/ OT (proposed) | 2.5% / 2.2% / 2.1%		| 0.424/0.487/0.508 |
> | FM w/ diffusion | 	   76.0% / 3.1% / 2.6%	| 0.066/0.344/0.478 |
> | SM w/ diffusion | 	   73.3% / 17.4% / 5.1%	| 0.062/0.176/0.349 |
>
> Exp 2: Training for 150K updates; inference with 8/16/32/64 NFEs. **We see that FM w/ OT can produce good results with just 8 NFEs, while FM w/ diff requires at least 16 NFEs and SM w/ diff requires over 64 NFEs.**
>
> | Model   | ZS-TTS WER (NFE=8/16/32/64) | ZS-TTS SIM-o (NFE=8/16/32/64) |
> | -------- | ------- | ------- |
> | FM w/ OT (proposed) | 2.4% / 2.2% / 2.2% / 2.1%		| 0.410/0.481/0.503/0.508 |
> | FM w/ diffusion | 	   11.5% / 3.0% / 2.7% / 2.6%		| 0.171/0.359/0.447/0.478 |
> | SM w/ diffusion | 	   94.5% / 42.3% / 11.5% / 5.1%	| 0.054/0.076/0.218/0.349 |
>
>
>
> Results on other setups show the same trend (all inference NFE and train steps combinations on ZS-TTS and diverse speech generation). We will add to the appendix in the final revision
>
> ---
>
> Given the positive feedback from the reviewers on novelty, potential impact, and our efforts in addressing the concerns raised, we feel that the scores do not accurately reflect on our work. We kindly request the reviewer to consider revising these scores to better align with the feedback provided. If there are specific areas where we can further improve to justify a higher score, we would greatly appreciate additional insights or guidance.

---

> > ### Comment · Reviewer_MQeA · 2023-08-21
> > **Thanks for your response**
> >
> > Thanks for your effort. The authors' response addressed my concerns about the fair experiments and they conducted ablation studies I suggest. Specifically, the ablation study for FM w/OT shows the effectiveness of the proposed method.
> >
> > Lastly, I would request to add the inference speed for each model in the final revision.
> >
> > Thanks. I will increase the score from 2 to 5.

---

> > > ### Comment · Reviewer_MQeA · 2023-08-22
> > > **Why do the authors mention the review on Twitter?**
> > >
> > > It seems that one of the authors mentioned "this review should be desk-rejected" on Twitter. Why???
> > >
> > > In the submitted manuscripts, they did not conduct the evaluation fairly and they should have conducted more ablation study to demonstrate the effectiveness of proposed methods. That is why I suggest additional experiments and the authors even conducted more experiments to address my concerns during discussion stage. However, I saw that one of the authors sniped my review by mentioning review may should be desk-rejected. I feel so bad in this situation.
> > >
> > > You should address the weakness of your paper in this rebuttal, not on Twitter.

---

> > > > ### Comment · Area_Chair_LxNb · 2023-08-22
> > > >
> > > > Thank you for bringing this issue to my attention. I will discuss it with SAC.

---

### Official Review · Reviewer_Peh7 · 2023-07-03

**Soundness:** 4 excellent
**Presentation:** 4 excellent
**Contribution:** 4 excellent
**Rating:** 8
**Confidence:** 4

**Summary:**

The paper proposes voicebox, a text-guided generative model for speech at scale. By abstracting many speech tasks into speech infilling tasks, voicebox is able to conduct zero-shot text to-speech synthesis, noise removal, content editing, style conversion, and diverse sample generation in monolingual or crosslingual scenario. Experiments show that equipped with the advanced technology such as CNF and classifier-free guidance, voicebox show better performance than baseline methods on many tasks,

**Strengths:**

1. The paper abstracts many different speech tasks into speech infilling task.
2. The paper scales up the model to 50K hours of speech and multilingual scenario, where the model is able to perform in-context learning for speech style.
3. The proposed method is well-studied on many tasks including zero-shot TTS, speech enhancement, speech editing and style generation. The experiments on ASR also show the potential value of proposed method.

**Weaknesses:**

About the experiments in Section 5.2, it is unfair to compare VB with speech enhancement model since VB can have access to the text of noised speech, which significantly help the model to generate speech with lower WER. Also, the text of noised speech is usually not accessible in the conventional formulation of speech enhancement.

**Questions:**

1. As Table 5 shows, VB is able to generate speech with different styles. Can this feature be used in Table 6? To be more specific, if VB synthesize more than one sample per text from the Librispeech training set, can the corresponding WER be further reduced?

2. If there is any insight for using Continuous Normalizing Flows to modeling speech infilling rather than other generative models such as diffusion models?

**Limitations:**

The authors have discussed the limitations in conclusion.

---

> ### Author Rebuttal · Authors · 2023-08-08
>
> We thank the reviewer for their thoughtful feedback. We address common questions in Author Rebuttal above and other questions here. We will incorporate all feedback in the final version.
>
> **1. As Table 5 shows, VB is able to generate speech with different styles. Can this feature be used in Table 6? To be more specific, if VB synthesize more than one sample per text from the Librispeech training set, can the corresponding WER be further reduced?**
>
> **Yes.** We have conducted such experiments before using an earlier version of the model (trained on 1K hour, using a different forced aligner and word-position-independent phones, using regression duration model). We generated three copies of the Librispeech training set, trained 1 ASR model on each copy, and trained another model combining three copies. The WER of the first three ASR models are 5.34%/5.38%/5.54% on test-clean and 15.40%/15.79%/15.83% on test-other. The WER of the ASR is 4.43% and 12.37% when trained on the combination of the three copies. The results demonstrate the potential of using Voicebox for data augmentation and we plan to explore more in future work.
>
> **2. About the experiments in Section 5.2, it is unfair to compare VB with speech enhancement model since VB can have access to the text of noised speech, which significantly help the model to generate speech with lower WER. Also, the text of noised speech is usually not accessible in the conventional formulation of speech enhancement.**
>
> We have noted in text that “It should be noted that A3T and Voicebox utilize transcript and location of the noise while Demucs does not.” We will revise this text to emphasize this difference more
>
> We agree that this is an unconventional setup for speech enhancement where transcript is usually unavailable. That said, we believe it is still a worthy application as transcripts can still be available in some scenarios. For example, it can be used when one records a scripted speech or when a robust audio-visual speech recognition model is available, which can transcribe accurately even at a very low SNR.

---

### Official Review · Reviewer_pReF · 2023-07-03

**Soundness:** 3 good
**Presentation:** 3 good
**Contribution:** 3 good
**Rating:** 5
**Confidence:** 4

**Summary:**

This paper proposed Voicebox, a non-autoregressive flow-matching model designed to infill speech by leveraging given audio context and text. Notably, Voicebox capitalizes on a substantial amount of data, consisting of 50,000 hours of speech, which contributes to its impressive performance across various speech generation tasks. The model demonstrates notable capabilities in generating coherent and high-quality speech outputs.

**Strengths:**

1. The paper exhibits a comprehensive and well-executed evaluation, yielding impressive experimental results.
2. The authors have effectively presented their work with clear and accessible writing, ensuring ease of understanding for readers.
3. The proposed framework showcases a high level of flexibility, enabling its application to various speech generation tasks and settings.

**Weaknesses:**

1. The demos provided in the supplementary materials are not as satisfying as described in the paper. For example, issues can be found in the perceived speaker similarity in the zero-tts task and the quality of the articulation position of the mask. It would be beneficial if the authors engage in further discussions regarding these phenomena.
2. Given that speech encompasses various components (such as prosody, content, timbre, and noise), it would be better to have a more comprehensive discussion on how Voicebox specifically handles these different aspects.

**Questions:**

It is unclear from the paper what motivated the authors to choose the flow-matching model as the audio model.

**Limitations:**

Authors are encouraged to add some discussions about the potential negative social impact.

---

> ### Author Rebuttal · Authors · 2023-08-08
>
> We thank the reviewer for their thoughtful feedback. We address common questions in Author Rebuttal above and other questions here. We will incorporate all feedback in the final version.
>
> **1. The demos provided in the supplementary materials are not as satisfying as described in the paper. For example, issues can be found in the perceived speaker similarity in the zero-tts task and the quality of the articulation position of the mask. It would be beneficial if the authors engage in further discussions regarding these phenomena.**
>
> All the samples presented in the supplementary material use audio prompts from internally recruited volunteers, who recorded samples from their own devices and provided consents for the processing and sharing samples online. Moreover, many of these speakers are non-native and with accents. These audio would be more out-of-domain relative to the training data, which may contribute to the perceptually lower similarity. VALL-E has observed similar results when tested on VCTK which is out-of-domain relative to their training data (Librivox). We will include audio samples for comparison with VALL-E if we can obtain consents for the prompts they used.
>
> In terms of the quality of the articulation position of the mask, we are unsure which samples have the issue reviewers mention. We would be more than happy to discuss if the reviewer could provide more details!
>
> **2. Given that speech encompasses various components (such as prosody, content, timbre, and noise), it would be better to have a more comprehensive discussion on how Voicebox specifically handles these different aspects.**
>
> Voicebox decouples speech into textual content and audio style, where audio style encompasses everything other than textual content. To generate an audio sample, the textual content is specified by the phone transcript, and the audio style is specified through the surrounding audio (audio context).
>
> Voicebox does not differentiate different aspects of audio style (voice, noise, emotion, etc) and does not need such labels either. Through learning to infill from large quantities of data, Voicebox learns that these attributes tend to be consistent within an utterance and it can infer the audio style of the target given the context.
>
> We can take the first “Transient Noise Removal” sample on the demo page as an example (transcript starts with “in zero weather in mid-winter…”). “Voicebox Input” audio shows the audio context input to the model. From it we hear a feminine voice speaking calmly, and there is static noise that is particularly noticeable when the speaker speaks. We can find the same audio style (low static noise, voice, slow pace, calm emotion) from the generated segments (“to a great depth below the surface when in driving over the”) presented in “Voicebox Output” audio.
>
> **3. Authors are encouraged to add some discussions about the potential negative social impact.**
>
> We have included some discussion of potential negative social impact and mitigation in Section 6. We expanded the discussion in the thread above for “Ethics Review” and will incorporate those in the final version.

---

### Official Review · Reviewer_K4NY · 2023-07-06

**Soundness:** 4 excellent
**Presentation:** 4 excellent
**Contribution:** 4 excellent
**Rating:** 7
**Confidence:** 5

**Summary:**

This paper proposes VoiceBox, a speech infilling model based on a flow-matching generative model. VoiceBox is trained to fill in masked speech based on unmasked speech and given text, and it can perform various tasks depending on how the mask is applied to the speech during inference. VoiceBox allows for speech editing by masking the speech corresponding to the text portion that needs to be edited, and it can also perform zero-shot TTS by infilling the speech for the desired text. By masking the entire speech during generation, the model can generate voices from various speakers in a speaker-unconditional manner for a given sentence. This paper demonstrates significantly improved performance compared to existing models in various tasks and even shows that a generative model for speech can aid in performance improvement in speech recognition tasks through diverse speech generation.

**Strengths:**

* This paper demonstrates the versatility of VoiceBox across various tasks while achieving impressive performance.
* This paper demonstrates that utilizing synthesized speech generated by VoiceBox can improve the generalizability of ASR models, thereby showing its ability to effectively model the distribution of general speech.
* In the zero-shot TTS task, VoiceBox particularly outperforms the previous state-of-the-art model, VALL-E, by a significant margin.
* The use of a flow-matching generative model enables fast sampling.
* The extensive experimental results provide strong support for the advantages of the model.

**Weaknesses:**

* For zero-shot TTS and speech editing, both the duration model and audio model require prompting with the transcript of the reference speech as well as the duration per phoneme. This necessitates the use of MFA (Montreal Forced Aligner) during inference.


**Questions:**

* Despite VALL-E already demonstrating impressive zero-shot TTS performance, VoiceBox appears to have excessively higher SIM-o and SIM-r scores. It is difficult to determine that VoiceBox's zero-shot TTS samples are significantly better than VALL-E when listening to only the VoiceBox samples in the demo. For a one-to-one comparison with VALL-E, would it be possible to provide VoiceBox samples generated with the same prompt as VALL-E's demo samples?

* In order to calculate SIM-r as mentioned in the paper, were both the reference speech prompt and all generated samples encoded and decoded using Encodec? If calculated differently, please provide an explanation. Additionally, when measuring SIM-o or SIM-r with the reference speech, was the similarity measured with the speech prompt only or with the entire reference speech?

*

**Limitations:**

This paper provides additional experiments on a classifier designed to detect potential misuse in order to mitigate such risks.

---

> ### Author Rebuttal · Authors · 2023-08-08
>
> We thank the reviewer for their thoughtful feedback. We address common questions in Author Rebuttal above and other questions here. We will incorporate all feedback in the final version.
>
> **1. In order to calculate SIM-r as mentioned in the paper, were both the reference speech prompt and all generated samples encoded and decoded using Encodec?  If calculated differently, please provide an explanation.**
>
> **No.** SIM-r aims to measure the similarity with respect to the audio feature target a generative model predicts. For Voicebox, we compute the mel spectrogram of the ground truth audio and decode the mel spectrogram into a waveform using HiFi-GAN to create the reference speech. We decode Voicebox output mel spectrogram into a waveform using HiFi-GAN to create the generated speech.
>
> SIM-r serves as the upper bound for the audio model given the selected audio feature (Encodec code for VALL-E / mel spectrogram for Voicebox). We noted in Section 4 that this number is not comparable when two models use different audio features or vocoders. Hence, we focus on SIM-orig and argue that SIM-orig should be used for comparison across papers.
>
> **2. when measuring SIM-o or SIM-r with the reference speech, was the similarity measured with the speech prompt only or with the entire reference speech?**
>
> When measuring SIM-{r,o}, the reference is the speech prompt. We confirmed this is the same protocol VALL-E used through personal communication with the authors.
>
> **3. Would it be possible to provide VoiceBox samples generated with the same prompt as VALL-E's demo samples?**
>
> The rebuttal period does not allow uploading audio samples, and we would also need explicit consent from a speaker to put samples resembling that speaker online. We will include them in the final demo if we could obtain such consents. We also note that all the samples presented in the supplementary material use audio prompts from internally recruited volunteers, who recorded audio from their own devices. These audio would likely be more out-of-domain.
>
> **4. For zero-shot TTS and speech editing, both the duration model and audio model require prompting with the transcript of the reference speech as well as the duration per phoneme. This necessitates the use of MFA (Montreal Forced Aligner) during inference.**
>
> We will add discussion on this in the main paper if space permits or in the appendix. We also note that for speech editing forced aligner is always needed during inference to identify the location of the source word(s) to be deleted/replaced, regardless of whether a forced aligner is used during training. In that regard the inference requirement of Voicebox is the same as prior work.
>
> To further resolve the limitation, there are two potential solutions to address the limitation in the future work. **First, we could use a mix of phonetic and self-supervised learning (SSL) units (e.g., HuBERT units) as the content representation similar to [1].** Specifically, the frame-level phone units corresponding to audio context can be replaced with SSL units, such that transcript of the context is not needed and duration of the SSL units can be easily derived since SSL units are originally at the frame level. **Second, we could explore using a separate encoder for audio context**, which duration model and audio model attend to through cross attention.
>
> [1] Fong, Jason, et al. "Speech Audio Corrector: using speech from non-target speakers for one-off correction of mispronunciations in grapheme-input text-to-speech." Interspeech’22

---

> > ### Comment · Reviewer_K4NY · 2023-08-18
> >
> > Thank you for addressing the concerns through the rebuttal. The additional two experiments effectively demonstrate the importance of data size for Voicebox and the motivation for using flow matching over the diffusion model. I believe this will be helpful for the readers, and it would be beneficial if you could incorporate this into the paper. Overall, I will keep my original score of 7.

---

### Official Review · Reviewer_F9C6 · 2023-07-11

**Soundness:** 3 good
**Presentation:** 3 good
**Contribution:** 3 good
**Rating:** 6
**Confidence:** 5

**Summary:**

The authors present Voicebox, the text-guided generative model for speech at scale. Voicebox is a non-autoregressive flow-matching model trained to infill speech, given audio context and text. Voicebox can be used for mono or cross-lingual zero-shot text-to-speech synthesis, noise removal, content editing, style conversion, and diverse sample generation.

**Strengths:**

It is interesting that Voicbox performs in-context learning via the masking strategy. Besides, the authors train the model in extensive data and present the SOTA results in several downstream tasks.

**Weaknesses:**

1. One of the weaknesses should be novelty. It sounds like the combination of masked-based A3T and flow-based speech synthesis model. Besides, duration prediction models and classifier-free guidance are not new.
2. Evaluation. Firstly, why present a portion of MOS in Table 2? VITS-LJ and VITS-VCTK are compared as the baseline in the LibriTTS test-other set, but why not train in the LibriTTS dataset instead? The differences in data amount could encounter unfair comparison.
3. Unclear presentation. Voicebox still needs to provide a clear illustration. Do you need tags for different task inferences? A3T finds that it is challenging to generate speech given full-mask samples, and what do you find is the most suitable proportion of masked and unmasked regions?

**Questions:**

1. It could be more natural to use diffusion models since the masking strategy is similar to adding noise, so what is your consideration in using flow-matching models?
2. Citations [42] and [43] are the same

**Limitations:**

/

---

> ### Author Rebuttal · Authors · 2023-08-08
>
> We thank the reviewer for their thoughtful feedback. We address common questions in Author Rebuttal above and other questions here. We will incorporate all feedback in the final version.
>
> **1. Do you need tags for different task inferences?**
>
> **No.** Voicebox performs different tasks by preparing input differently, which we illustrated in Figure 1. We have added refined illustrations in the supplementary PDF for the rebuttal (Figure 1-3).
>
> **2. Why present a portion of MOS in Table 2?**
>
> **VALL-E is not publicly available** and there are only 8 samples from their demo page, which are not sufficient for MOS studies. We did not present A3T MOS because **the performances of A3T on WER and SIM-o are very bad**, and the MOS scores are expected to be very bad after we listen to a number of samples.
>
> **3. A3T finds that it is challenging to generate speech given full-mask samples, and what do you find is the most suitable proportion of masked and unmasked regions?**
>
> A3T finds it challenging because it assumes the mapping between the input (text and audio context) and the output (target audio) is deterministic, as discussed in paragraph 3 Section 2 and the response to common comment 1 above. In particular, it struggles more as the duration to infill becomes longer (the distribution of possible speech becomes broader and less described by the mean targeted by regression).
>
> In contrast, Voicebox addresses the issue by modeling with a CNF model, which models the full distribution over possible speech, and not just the mean like for a regression model. Hence, Voicebox is capable of infilling audio of any length.
>
> During training, we mask the entire audio with 30% probability, and with 70% probability masking a contiguous chunk that is [70%, 100%] the length of the entire audio. During inference, it depends on the task and the input. For zero-shot TTS, the masked length is the length of predicted duration of the target transcript. For denoising, we explore infilling 50% (Table 3) and 30%/70% (Table B4 in Appendix). For diverse speech sampling (Section 5.3), it is 100% masked.
>
> **4. Citations [42] and [43] are the same**
>
> We will fix that. Thank you.

---

> > ### Comment · Area_Chair_LxNb · 2023-08-18
> >
> > Thank you for answering the questions and clarifying the details.

---

### Author Rebuttal · Authors · 2023-08-08

We thank the reviewers for their thoughtful feedback. We are glad that they found Voicebox is highly versatile and can perform many different tasks (**FPC6, K4NY, pReF, Peh7**) by abstracting them into speech infilling (**Peh7**). We are encouraged that they found Voicebox is well-evaluated (**K4NY, pReF, Peh7**), scales effectively to 50K hours and multilingual setups (**Peh7, F9C6, MQeA**), shows SoTA performance (**F9C6, K4NY, pReF, Peh7**), enables faster inference through flow-matching with OT  (**K4NY**), and has the potential as a data generator for training other models  (**K4NY, Peh7**). We are also pleased that reviewer **pReF** found the paper easy to follow. We address common questions here and individual ones in separate threads. We will incorporate all feedback in the final version.

**Comment 1: Novelty (MQeA, F9C6)**
> **MQeA**: “The concept to train the model by infilling speech given audio and text has been already presented in many works.” **F9C6**: “It sounds like the combination of masked-based A3T and flow-based speech synthesis model”

**We highlight that no prior speech infilling models has attempted to or is capable of generalizing to as many tasks in the context of generative modeling as Voicebox does.** The goal of this paper is to build a generalist speech generative model that can solve many tasks without fine-tuning, just like LLMs solving many NLP tasks. Below we discuss the key novelties:

1. We show that speech infilling subsumes many tasks which have not been presented before. While speech infilling models exist, they struggle to generate long or diverse samples, because they assume deterministic input/output mapping and formulate it as a regression task.
2. We show that NAR flow-matching overcomes deterministic mapping and speeds up training and inference. See Table B3 (our reduced setup with 150K steps achieves better WER/spk similarity than Vall-E). See Fig. 2 for inference time comparison.
3, Voicebox enables unprecedented scaling and a single multilingual generative model. We added experiments in the rebuttal PDF Table 1 to show the benefit.

**Comment 2: Why flow-matching w/ OT path (F9C6, pReF, Peh7, MQeA)**
> **F9C6**: “use diffusion models since the masking strategy is similar to adding noise.” **pReF**: “what motivated the authors to choose the flow-matching model.” **Peh7**: “insight for using CNF to model speech infilling rather than other generative models such as diffusion models?” **MQeA**: “compare the conditional normalizing flow with a diffusion-based model.”

**To generalize speech infilling, any powerful non-autoregressive generative models, including diffusion models, should work. We chose flow-matching (FM) with optimal transport (OT) path because [1] showed that FM w/ OT > FM w/ diffusion > score-matching (SM) w/ diffusion (the typical diffusion model) on training speed and inference compute-quality trade-off.** See the comparison in Table 1 and Fig 4-7 in [1].

FM and SM models are in fact similar. Both transform a noise distribution to the data distribution, and learn to predict the gradient (score and flow, respectively) given a time step and a noisy sample at that time step. Diffusion and OT are simply two different paths (with the same initial and final marginal distribution). A path dictates how a sample transforms between initial and final distribution. As shown in Figure 3, Lipman et al. (2023), OT corresponds to a “simpler path” with constant speed and direction compared to diffusion, which the authors argue is easier to learn (faster training) and integration can be estimated more accurately with fewer steps (faster inference), leading to better empirical results.

[1] Lipman et al. "Flow matching for generative modeling." ICLR’23

**Comment 3: Baselines (F9C6, MQeA)**
> **F9C6**: “why not train [VITS] in the LibriTTS dataset?” **MQeA**: “YourTTS is not a good TTS model for zero-shot TTS models. The audio quality of YourTTS is bad. I recommend training the VITS with the reference encoder you used and the same configuration.”

**YourTTS is exactly a VITS model trained on LibriTTS (and a few other datasets) with a reference encoder.** We have included such comparisons in Table 5 which reviewer F9C6 asked for. In addition, the Voicebox with a reduced setup (12 layers, trained on 1K hours) in Table B3 in the appendix still shows superior performance than VITS/YourTTS. Moreover, we trained a VITS on LibriSpeech without a reference encoder. It leads to much worse results (16.64% WER and 311.75 FSD for Table 5)

**Unfortunately, we cannot train VITS with same reference encoder because Voicebox doesn't have an explicit encoder**. Voicebox is a decoder only model taking text, masked audio, noisy audio) as input to predict flow. Also, given YourTTS is exactly VITS with a reference encoder trained on data similar to ours, we are not sure why the suggested experiment would lead to better results than YourTTS.

> **MQeA**: “all models should be trained with the same dataset.” **MQeA**: “transferring the voice style from a reference audio, not speaker ID, is utilized in many works. [...] the same transferring method is used for each model.” **MQeA**: “more comparisons with the recently proposed TTS models (not YourTTS, not Vall-E)”

Given that this work targets generalist speech generative model with unprecedented scaling, we believe **YourTTS is an appropriate baseline** because it is recent (ICML’22), open-sourced, achieves SoTA prior to VALL-E on zero-shot TTS, and is trained on in-the-wild data which enables better generalization than those trained on VCTK/LJ.

Furthermore, **we compared with most related models trained with the same dataset and using the same transferring method**, which are VALL-E and A3T-style models (Section B.3 in the appendix). We could better address the comments if reviewer could kindly clarify **why these are not appropriate baselines**, and **what zero-shot TTS models we should compare with and why.**

---

> ### Author Response · Authors · 2023-08-18
> **Two additional experiments to address Reviewer MQeA's new comments**
>
> We thank all the reviewers again for spending time carefully reading our responses. We have added two more experiments to address Reviewer MQeA’s follow-up comments, and would like to highlight the additional experiments in the common thread. Please see the details below.
>
> ---
>
> **1. We present a new experiment here to address the concern on the comparison with YourTTS in terms of data.** We trained Voicebox on LibriTTS with the reduced setup described in appendix Section B3, which is strictly a subset of what YourTTS is trained on (LibriTTS, VCTK, PT, FR SCL). Results suggest that Voicebox are still significantly better
>
> | Model   | ZS-TTS WER | ZS-TTS SIM-o |
> | -------- | ------- | ------- |
> | Voicebox trained on LibriTTS | 2.1% | 0.579 |
> | YourTTS | 7.7% | 0.337 |
>
> ---
>
> **2. We present new experiments comparing FM w/ OT, FM w/ diffusion, and SM w/ diffusion. We confirm empirically that FM w/ OT indeed has significantly better training and inference efficiency.** We adopt the same ablation setup as Appendix Section B.3 (with loss computed on all frames and a smaller learning rate, 1e-4, to ensure convergence for all three methods). We vary the number of training and inference steps and show experimental results below:
>
>
> Exp 1: Training for 50K / 100K / 150K updates; Inference with 64 NFEs. **We see FM w/ OT achieves the best performance with 100K training steps, and even outperforms SM w/ diff using only 50K updates.**
> | Model   | ZS-TTS WER (upd=50K/100K/150K) | ZS-TTS SIM-o (upd=50K/100K/150K)  |
> | -------- | ------- | ------- |
> | FM w/ OT (proposed) | 2.5% / 2.2% / 2.1%		| 0.424/0.487/0.508 |
> | FM w/ diffusion | 	   76.0% / 3.1% / 2.6%	| 0.066/0.344/0.478 |
> | SM w/ diffusion | 	   73.3% / 17.4% / 5.1%	| 0.062/0.176/0.349 |
>
> Exp 2: Training for 150K updates; inference with 8/16/32/64 NFEs. **We see that FM w/ OT can produce good results with just 8 NFEs, while FM w/ diff requires at least 16 NFEs and SM w/ diff requires over 64 NFEs.**
> | Model   | ZS-TTS WER (NFE=8/16/32/64) | ZS-TTS SIM-o (NFE=8/16/32/64) |
> | -------- | ------- | ------- |
> | FM w/ OT (proposed) | 2.4% / 2.2% / 2.2% / 2.1%		| 0.410/0.481/0.503/0.508 |
> | FM w/ diffusion | 	   11.5% / 3.0% / 2.7% / 2.6%		| 0.171/0.359/0.447/0.478 |
> | SM w/ diffusion | 	   94.5% / 42.3% / 11.5% / 5.1%	| 0.054/0.076/0.218/0.349 |
>
>
> Results on other setups show the same trend (all inference NFE and train steps combinations on ZS-TTS and diverse speech generation). We will add to the appendix in the final revision

---

### Decision · Program_Chairs · 2023-09-21

**Decision:**

Accept (poster)

**Comment:**

This paper proposes a non-autoregressive flow-matching model for generating speech given audio context and text. The model was trained using in-context learning with a large-scale dataset of over 50,000 hours of speech. It also provides a comprehensive comparison with the state-of-the-art VALL-E and other models.

The paper was thoroughly discussed and ultimately received acceptance from all reviewers, with reviewer MQeA particularly changing their score from 2 to 5, leading to an “Accept” decision.